

# Global evaluation of nutrient enabled version land surface model ORCHIDEE-CNP v1.2 (r5986)

Yan Sun[1], Daniel S Goll[1,2], Jinfeng Chang[3], Philippe Ciais[1], Betrand Guenet[1,4], Julian Helfenstein[5], Yuanyuan Huang[1,6], Ronny Lauerwald[1,7], Fabienne Maignan[1], Victoria Naipal[1,8], Yilong Wang[1,9], Hui Yang[1], Haicheng Zhang[1,7]

[1]Laboratoire des Sciences du Climat et de l'Environnement/IPSL, CEA-CNRS-UVSQ, Université Paris-Saclay, Gif sur Yvette, 91191, France.
[2] Department of Geography, University of Augsburg, Augsburg, Germany
[3] Ecosystems Services and Management Program, International Institute for Applied Systems Analysis (IIASA), Schlossplatz 1, A-2361 Laxenburg, Austria
[4]Laboratoire de Géologie, UMR 8538, Ecole Normale Supérieure, PSL Research University, CNRS, Paris, France
[5]Agroecology and Environment, Agroscope, Reckenholzstrasse 191, 8046 Zurich, Switzerland
[6] CSIRO Oceans and Atmosphere, Aspendale 3195, Australia
[7] Department Geoscience, Environment & Society, Universite libre de Bruxelles, 1050 Bruxelles, Belgium
[8] Department of Geography, Ludwig-Maximilian University, Munich, Germany
[9] Key Laboratory of Land Surface Pattern and Simulation, Institute of Geographical Sciences and Natural Resources Research, Chinese Academy of Sciences, Beijing, China

*Correspondence to*: Yan SUN (ysun@lsce.ipsl.fr); Daniel S. Goll (dsgoll123@gmail.com)

**Abstract.** The availability of phosphorus (P) and nitrogen (N) constrain the ability of ecosystems to use resources such as light, water and carbon. In turn, nutrients impact the distribution of productivity, ecosystem carbon turnovers and their net exchange of $CO_2$ with the atmosphere in response to variation of environmental conditions both in space and in time. In this study, we evaluated the performance of the global version of the land surface model ORCHIDEE-CNP (v1.2) which explicitly simulates N and P biogeochemistry in terrestrial ecosystems coupled with carbon, water and energy transfers. We used data from remote-sensing, ground-based measurement networks and ecological databases. Components of the N and P cycle at different levels of aggregation (from local to global) are in good agreement with data-driven estimates. When integrated for the period 1850 to 2017 forced with variable climate, rising $CO_2$ and land use change, we show that ORCHIDEE-CNP underestimates the land carbon sink in the North Hemisphere (NH) during the recent decades, despite an a priori realistic GPP response to rising $CO_2$. This result suggests either that other processes than $CO_2$ fertilization which are omitted in ORCHIDEE-CNP, such as changes in biomass turnover, are predominant drivers of the northern land sink, and/or that the model parameterizations produce too strict emerging nutrient limitations on biomass growth in northern areas. In line with the latter, we identified biases in the simulated large-scale patterns of leaf and soil stoichiometry and plant P use efficiency pointing towards a too severe P limitations towards the poles. Based on our analysis of ecosystem resource use efficiencies and nutrient cycling, we propose ways to address the model biases by giving priority to better representing processes of soil organic P mineralization and soil inorganic P transformation, followed by refining the biomass

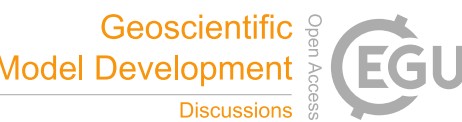

production efficiency under increasing atmospheric $CO_2$, phenology dynamics and canopy light
absorption.

## 1 Introduction

Nitrogen (N) and phosphorus (P) are key macronutrients that control metabolic processes and plant
growth and constrain ecosystem-level productivity (Elser et al., 2007; Norby et al., 2010; Cleveland et
al., 2013). The amount and stability of soil carbon (C) stock is also affected by N and P through their
regulating role in the mineralization of litter and soil organic matter (Gärdenäs et al., 2011; Melillo et al.,
2011). The availability of N and P is likely to limit future carbon storage under climate change and
rising atmospheric $CO_2$. Empirical stoichiometry observations were applied in the posteriori estimates
of future carbon storage from land surface models (LSM) lacking an explicit simulation of N and P
biogeochemistry, which pointed out consistently in this direction of future carbon storage (Hungate et
al., 2003; Wang and Houlton, 2009; Zaehle et al., 2015; Wieder et al., 2015). Nevertheless, this
approach has large uncertainties (Penuelas et al., 2013; Sun et al., 2017) and relies on unproven
assumptions (Brovkin and Goll, 2015).

An alternative is to represent directly the complex interactions between N, P and carbon in a LSM.
Several LSMs incorporated different parameterizations of N interactions (e.g. Thornton et al., 2007;
Zaehle et al., 2014) but very few global models have included P interactions. The few models accounted
for P limitation in plant growth showed that P availability limit primary productivity and carbon stocks
on highly weathered soils of the tropics (Wang et al., 2010; Yang et al., 2014) and one study also
suggested that P limitations could also occur in the e northern hemisphere in the near future (Goll et al.,
2012). Model representations of P interactions are highly uncertain since the critical processes are
poorly constrained by current observational data. In particular, the desorption of P from soil minerals
surface and the enhancement of P availability for plants by phosphatase enzymes secreted by plant roots
and microbes were identified to be critical but poorly constrained (Fleischer et al., 2019).

Previous studies (Wang et al., 2010; Goll et al., 2012; Yang et al., 2014; Thum et al., 2019) have
suggested that the inclusion of the phosphorus cycle improves model performances with regard to
reproducing observed C fluxes. But adding new and uncertain P-related processes does not grant an
automatic improvement of a land surface model in general. First, more (nutrient-related) equations with
more uncertain parameters can result in less robust predictions. Second, models ignoring nutrients were
often calibrated on available carbon data, so that a new model with nutrients inevitably needs a
parameter recalibration to reach the similar performances than the same model without nutrients. Third,
for evaluating a large-scale model resolving both nutrient and carbon biogeochemistry, one should look
for specific nutrient related datasets which are more scarce than classical biomass, productivity, soil
carbon data used for benchmarking carbon only models.

The evaluation for N and P together with carbon cycling in global LSM models remains very limited
(Wang et al., 2010; Goll et al., 2012) but recent ground-based measurements and ecological datasets
offer the opportunity to make progress. With recent meta-analysis of site-level nutrient fertilization
experiments (e.g. Yuan and Chen, 2015; Wright, 2019), data-driven assimilation schemes to constrain



nutrient budgets (Wang et al., 2018), new knowledge about the critical P-processes of sorption (Helfenstein et al., 2018; 2020) and phosphatase-mediated mineralization (Sun et al., 2020), global datasets of leaf nutrient content (Butler et al., 2017), and empirical constraints on the $CO_2$ fertilization
effect on land carbon storage (Terrer et al., 2019; Liu et al., 2019), a better evaluation of C, N, P models is feasible. In addition to direct comparison with nutrient datasets, it is also possible to diagnose emerging model responses in terms of ecosystem resource use efficiencies (RUE) and confront them to observations for identifying how ecosystems adjust and optimize nutrient, water, light, and carbon resource availabilities (Fernández-Martínez et al., 2014; Hodapp et al., 2019). In particular, modeled N
and P use efficiencies can be compared to observation-based estimates at ecosystem scale (Gill and Finzi, 2016) and at biome scale (Wang et al., 2018).

Here we evaluate the global cycles of C, N and P in the nutrient-enabled version of the LSM ORCHIDEE, ORCHIDEE-CNP (v1.2). The model has been previously evaluated for tropical sites (Goll et al., 2017a, 2018) and for coarse scale global carbon fluxes and stocks using the International Land
Model Benchmarking system iLAMB by e.g. Friedlingstein et al., (2019). The results from this evaluation showed a slightly worse performance for ORCHIDEE-CNP (v.1.2) than the carbon-only version of ORCHIDEE which has been extensively calibrated (Friedlingstein et al., 2019). In this study, we perform a detailed evaluation of ORCHIDEE-CNP focusing on four aspects: 1) spatial-temporal patterns of terrestrial C fluxes, 2) N and P fluxes and budgets at catchment, biome and global scale, 3)
large scale pattern of the stoichiometry of leaves and soils, 4) plant resource use efficiencies for light, water, C, N and P.

## 2 Model description

ORCHIDEE-CNP simulates the exchange of greenhouse gases (i.e. carbon dioxide, nitrous oxide), water and energy at the land surface and features a detailed representation of the root uptake of
dissolved N and P, the allocation of N and P among tissues, and the N and P turnover in litter and soil organic matter (Goll et al., 2017a, 2018) (Fig. 1). In this study, we present the first global application of the model and an evaluation against global carbon and nutrient datasets. The model version applied here is based on Goll et al. (2017a, 2018) and referred to as ORCHIDEE-CNP v1.2. Major modifications compared to v1.1 are described as follows (details can be found in the Text S1).

The original formulation of photosynthetic capacity in ORCHIDEE-CNP v1.1 assumed leaf N to be the sole regulator of leaf photosynthetic characteristic (Kattge et al., 2009). Here, we applied a new empirical function that relates photosynthetic capacity to both leaf N and P concentration based on data from 451 species from 83 different plant families (Ellsworth et al., in prep.). A priori and narrow plant functional type (PFT)-specific range of leaf C:N:P ratios that were prescribed in ORCHIDEE-CNP v1.1
are now given a larger range common to all PFTs (Table S1), allowing for the prediction of variation of leaf stoichiometry across climate and soil gradients, independently of the prescribed vegetation (PFT) map.

In ORCHIDEE-CNP v1.1, an empirical function, $f(T_{soil})$ was used to reduce biochemical mineralization and plant nutrient uptake at low soil temperature (Eq.5 in Goll et al., 2017a) which was adopted from



the N enabled version of ORCHIDEE (Zaehle and Friend, 2010) to avoid an unrealistic accumulation of N within plants when temperatures are low. We found that function was not needed when P uptake is accounted for and was thus removed. It should be noted that this temperature dependence is different from the one which describes the temperature dependence of SOM and litter decomposition. For grasslands and croplands, we implemented root dormancy which is triggered by drought or low
temperatures. During dormancy, root maintenance respiration is reduced by 90% following (Shane et al., 2009) but root acquisition of soil nutrients continues as long as root biomass exists (Malyshev and Henry, 2012). It should be noted that total root loss can occur for extremely long droughts or cold periods when maintenance respiration depletes root carbon.

Several parameters were re-calibrated, i.e. the coefficient relating maintenance respiration to biomass
and the leaf to sapwood ratio, or corrected in case of the turnover of sapwood for tropical evergreen broadleaf forest (TREBF) and tropical rain-green broadleaf forest (TRDBF) to achieve more realistic wood growth rates for those forests (not shown). We also adjusted the recycling efficiency of nutrients from root ($f^N_{trans,root}$, $f^P_{trans,root}$) and leaf ($f^N_{trans,leaf}$, $f^P_{trans,leaf}$) according to data compilations from Freschet et al. (2010) and Vergutz et al. (2012). The new values of these parameters and their sources
are given in Supplementary Information (Text S1).

### 3 Simulation protocol, forcing and evaluation datasets

#### 3.1 Simulation protocol and forcings

##### 3.1.1 Simulation setup

We performed a global simulation at $2^o$ x $2^o$ spatial resolution for the historical period (1700-2017)
adapting the TRENDY version 6 protocol (Sitch et al., 2015; Le Quéré et al., 2018). The simulation was performed using historical climate forcing, land cover changes and management (i.e. mineral fertilizer application, crop harvest, see 3.1.6), and atmospheric $CO_2$ concentrations (S3 type simulation). Prior to the historical simulation, we performed a model spin-up to equilibrate the C, N and P pools and fluxes (Appendix A) by forcing the model with cycled climate forcing of 1901-1920 and the land cover map
and land management corresponding to the year 1700. To benchmark ORCHIDEE-CNP v1.2, we performed the same simulation with ORCHIDEE revision 5375 which is identical for other processes but has no nutrient cycles parameterization.

##### 3.1.2 Meteorological data

The model was forced by CRU-JRA-55 meteorological data provided at a spatial resolution of $0.5^o$ x
$0.5^o$ and upscaled to a resolution of $2^o$ x $2^o$. This data comprises global 6-hourly climate forcing data providing observation-based temperature, precipitation, and incoming surface radiation. It is derived from Climatic Research Unit (CRU) TS3.1 monthly data (Harris et al., 2014) and the Japanese 55-year Reanalysis (JRA-55) data (Kobayashi et al., 2015), covering the period 1901 to 2017. This climate dataset was provided by the TRENDY-v6 model-intercomparison project (Le Quéré et al., 2018).



### 3.1.3 Land cover

The simulations were driven by the History Database of the Global Environment land use data set (HYDE v3.2; Klein Goldewijk et al., 2017a, b), which provides fractional data for natural vegetation, cropland and pasture with a spatial resolution of $0.5^o$ x $0.5^o$. The HYDE land use data set was developed
based on historical population, cropland and pasture statistics combined with satellite information in the last decade and specific agricultural land use allocation algorithms, and is available for the period 1700-2018. We translated the land cover data from HYDE into the spatial distribution of 15 Plant Functional Type (PFT) area fractions used in ORCHIDEE, following the cross-walking method by Poulter et al. (2011, 2015). An updated release of the historical land-use forcing data set Land-Use Harmonization
(LUH) v2 (http://luh.umd.edu/data.shtml; Hurtt et al., 2011) was applied to constrain the land-cover transitions between forest, grassland (combining pasture and natural grassland), and cropland during the period 1860–2017 using the backward natural land cover reconstruction method (BM3) of Peng et al. (2017). A fixed PFT map of year 1860 was used for years prior to 1860.

### 3.1.4 Soil datasets and soil map

ORCHIDEE-CNP v1.2 is forced by information on soil texture, pH, bulk density and soil types (Goll et al., 2017a). We used a global gridded map of three soil texture classes from Zobler (1986) to derive soil-texture-specific parameters for soil water capacity, hydraulic conductivity and thermal conductivity. We used global gridded data on bulk density from the Harmonized World Soil Database (HWSD,
30 FAO/IIASA/ISRIC/ISSCAS/JRC, 2012) and soil pH from International Geosphere-Biosphere
Programme Data Information System Soil Data (Global Soil Data Task Group, 2000). Soil pH forcing maps are needed to simulate the dynamics of $NH_3$ and $NH_4^+$ in soil in ORCHIDEE (Zaehle and Friend, 2010). We used a global gridded map with the dominant soil orders (following the USDA Soil Taxonomy) at $1^o$x$1^o$ resolution to derive soil order specific soil phosphorus sorption parameters (Goll et al., 2017a).

### 175 3.1.5 Lithology map and soil shielding

The release of P from chemical weathering of rocks is computed dynamically following Goll et al. (2017a) and depends on the lithology types and soil shielding (discontinuation of the active soil zone from the bedrock) (Hartmann et al., 2014). We used the global lithological map (GLiM) of Hartmann and Moosdorf (2012) upscaled to $1^o$ x $1^o$ resolution which accounts for the lithology fractional coverage
of 16 classes on a sub-grid scale. We also used a spatial explicit map of soil shielding on a $1^o$ x $1^o$ resolution (Hartmann et al., 2014).

### 3.1.6 Atmospheric N and P deposition

Global gridded monthly atmospheric N and P deposition during 1860-2017 was derived from a reconstruction based on the global aerosol chemistry–climate model LMDZ-INCA (Wang et al., 2017).
LMDZ-INCA was driven by emission data, which included sea salt and dust for P, primary biogenic aerosol particles for P, oceanic emissions for N ($NH_3$), vegetation emissions for N (NO), agricultural





activities (including fertilizer use and livestock) for N and fuel combustion for both N ($NO_y$ and $NH_x$) and P. Reconstructions for the years 1850, 1960, 1970, 1980, 1990, and each year from 1997 to 2013 were linearly interpolated to derive a time series for 1850-2013. For the period before 1850, we assumed N and P deposition rates of the year 1850. For the period after 2013, we assumed rates of the year 2013. In ORCHIDEE-CNP, atmospheric N and P deposition are added to the respective soil mineral N and P pools without considering interception by the canopy.

### 3.1.7 Mineral and manure N and P fertilization for croplands and pasture

For croplands, we used yearly gridded mineral N and P fertilizer application data from Lu and Tian (2017) available for the period 1960 to 2017. This dataset is based on national-level data of crop-specific fertilizer application amounts from the International Fertilizer Industry Association (IFA) and the FAO. N and P mineral fertilization between 1900 and 1959 were linearly extrapolated assuming that fertilizer applications for 1900 are zero, and that there were no N and P fertilizers applied before 1900. For pasture, we used global gridded datasets of mineral N fertilizer application rates from Lu and Tian (2017), developed by combining country-level statistics (FAO) and land use datasets (HYDE 3.2) (Xu et al., 2019). For both cropland and pasture, N and P in mineral fertilizer was assumed to go directly into soil mineral pools, where all mineral N fertilizer was assumed to be in the form of ammonium nitrate, that is half of N as ammonium ($NH_4^+$) and half as nitrate ($NO_3^-$).

Manure applications are also included as a model forcing, given their significant input contribution to agricultural soils. For cropland, we used gridded annual manure N application data for the period 1860–2014 from Zhang et al. (2017) compiled and downscaled based on country-specific annual livestock population data from FAOSTAT. For the period before 1860, we assumed N and P deposition rates of the year 1860. For pasture, we used global gridded datasets of N manure application rates from Lu and Tian (2017). The application of manure P in cropland and pasture was derived from manure N assuming a manure P:N ratio of 0.2. This ratio is a weighted value by the amount of manure N applied to soil and derived from ruminants (14.4 Tg N $yr^{-1}$) and monogastric animals (10.1 Tg N $yr^{-1}$) from FAOSTAT for the year 2000 with P:N ratios of 0.165 in ruminant manure (mean of 0.15-0.18 from Lun et al., 2018) and 0.26 in monogastric manure (mean of 0.24-0.28 from Lun et al. (2018)). For manure applied to cropland and pasture, we assumed a typical slurry application with 90% of N in the liquid part of the slurry (like urine) goes to soil $NH_4^+$ pool. For the solid part of the slurry, we assumed it goes to a litter pool with a C:N ratio of 10:1 following Soussana and Lemaire (2014).

Mineral and manure N and P fertilizers in cropland were applied at day of year (DOY) 120 for northern hemisphere (30$^o$N - 90$^o$N), at DOY 180 for tropical regions (30$^o$N - 30$^o$S), and at DOY 240 for southern hemisphere (30$^o$S - 90$^o$S).

### 3.2 Evaluation datasets

We evaluated the performances of ORCHIDEE-CNP v1.2 based on four major aspects (Fig. 1). Each dataset is summarized in Table 1 and described in detail in the Supplementary Information. All the





gridded datasets with high spatial resolutions (Table 1) were resampled to the 2° x 2° resolution of the
model output using area-weighted mean methods. Here we give a brief overview of the evaluation.
Firstly, we evaluated the spatiotemporal patterns of terrestrial C fluxes and pools of the two versions of
ORCHIDEE. This is necessary to assess how the inclusion of nutrients affects the simulated C cycle and
the terrestrial C sink from the historical perturbation of $CO_2$, climate, land use and nutrient deposition.
For this, we used observation-based products of GPP, NPP, biomass and soil carbon pools and
atmospheric inversions of the net land-atmosphere $CO_2$ flux excluding fossil fuel emissions (Table 1).
Secondly, we evaluated N and P fluxes and budgets on global, biome and catchment scale to assess the
extent to which ORCHIDEE-CNP v1.2 is able to reproduce the observed fluxes of these two nutrients.
For this, in the absence of global spatially resolved observations of N and P fluxes, we used the data-
driven reconstruction of steady state N and P fluxes established with data assimilation system Global
Observation-based Land-ecosystems Utilization Model of Carbon, Nitrogen and Phosphorus (GOLUM-
CNP) v1.0 (Wang et al., 2018) for different biomes (Table 1). We also used observation-based estimates
of soil N and P leaching at catchment scale from Mayorga et al. (2010). Thirdly, we evaluated the large
scale patterns of vegetation and soil N:P ratios which reflect the relative limitations of N vs. P on plants
and decomposers (Güsewell, 2004). For this, we used global compilations of site-level leaf
stoichiometry (McGroddy et al., 2004; Reich and Oleksyn, 2004; Kerkhoff et al., 2005) and soil
stoichiometry measurements (Tipping et al., 2016) and global maps of leaf N:P ratios upscaled from site
measurements using a machine-learning model (Butler et al., 2017) (Table 1). Finally, we evaluated
resource use efficiencies diagnosed from GPP for light, water, C, N and P on global and biome scales.
Considering that some variables required for the global evaluation are not directly given by evaluation
datasets, we illustrate below how we derived those variables from evaluation datasets (section 3.2.1 and
3.2.3) and how we correspond those variables in ORCHIDEE-CNP (section 3.2.2).

### 3.2.1 Data-driven estimate of net (NPP) and gross primary productivity (GPP)

Different data-driven maps of NPP and GPP based on remote sensing and climate datasets were used
(Table 1), thereby accounting for the uncertainty of each product as well as for the uncertainty from the
spread between different products. Uncertainties of each NPP and GPP product were derived according
to original publications. We used a 20% uncertainty of gridded GPP from Moderate Resolution Imaging
Spectroradiometer (MODIS) and Breathing Earth System Simulator (BESS) (Appendix C; Turner et al.,
2006; Jiang and Ryu, 2016) at 2° scale. This is a coarse extrapolation of uncertainty reported at grid-cell
scale, since none of these products reported spatial error covariance information allowing to up-scale
this uncertainty at 2° resolution. Further, for some products, uncertainty was defined as the bias against
local measurements (Turner et al., 2006) and for others by using different climate input fields (Table 1).
For Multi-Tree-Ensemble (MTE)-GPP (Table 1), we used the spread (1-sigma standard deviation) from
an ensemble of 25 members produced by different machine learning methods (Jung et al., 2009). For
MODIS-NPP (Table 1), we used a 19% uncertainty as assessed by Turner et al. (2006). For BETHY-
NPP we do not have an uncertainty (Tum et al., 2016). For Global Inventory Modeling and Mapping
Studies (GIMMS)-NPP (Table 1), we used the variance of three sets of products (Table 1) based on
different climate datasets (Smith et al., 2016).





### 3.2.2 Losses of N and P from soils to river catchments

We used the export of N and P from terrestrial systems from the Global Nutrient Export from Water-Sheds(GlobalNEWS2) model (Table 1). This model gives N and P loading to rivers by surface runoff and drainage based on a mass-balance approach (Mayorga et al., 2010). As ORCHIDEE-CNP lacks a representation of nutrient soil infiltration and groundwater networks, we had to approximate N and P load from ORCHIDEE output variables. Since most of dissolved N in groundwater enters the river

network via drainage (Bouwman et al., 2013a), we extracted annual N exported via drainage and surface runoff from the ORCHIDEE-CNP output to be compared with the GlobalNEWS2 load rates ($N_{load}$) on catchments scale:

$$N_{load} = N_{runoff} + N_{drainage} \,, \tag{1}$$

where $N_{runoff}$ is the simulated annual N flux lost by surface runoff, and $N_{drainage}$ the annual N flux

lost by drainage both in kgN km$^{-2}$ yr$^{-1}$.

Dissolved P which drains in deeper soil layers ($P_{dra}$) is mostly adsorbed by soil minerals and only a minor fraction enters the groundwater. Thus, the we extracted from the ORCHIDEE-CNP output the annual P loss from soils via surface runoff only (kg P km$^{-2}$ yr$^{-1}$) to be compared with the GlobalNEWS2 load rates ($P_{load}$):

$$P_{load} = P_{runoff} \,, \tag{2}$$

### 3.2.3 Uncertainties in observation-based resource use efficiencies

The definition of resource use efficiencies is explained in section 4.4. Observation-based light use efficiency (LUE) was calculated using MTE-GPP, downward shortwave radiation from CRUJRA, and fraction of Absorbed Photosynthetically Active Radiation (fAPAR) from the Global SeaWiFS Level-3

data (Gobron et al., 2006a, b). Uncertainty was derived from 25 ensemble members of MTE-GPP. Observation-based water use efficiency (WUE) was calculated as the ratio between MTE-GPP and MTE-ET (Table 1); its uncertainties calculated using a Monte-Carlo resampling procedure in which 25 different members of GPP and ET were randomly selected. Observation-based carbon use efficiency (CUE) was calculated from the ratio of MODIS-NPP to MODIS-GPP. It should be noted that MODIS-

NPP is based from a calibrated version of the BIOME-BGC model (Turner et al., 2006) so that CUE is not strictly speaking an observation-based quantity. CUE uncertainties were calculated using a Monte-Carlo method given a 20% and 19% uncertainty for MODIS GPP and NPP products at 2° resolution, respectively.



## 4. Results

### 4.1 Evaluation of carbon cycling variables

In the following, we analyze the performance of ORCHIDEE-CNP v1.2 and ORCHIDEE without nutrient cycles with respect to the spatiotemporal patterns of GPP, NPP and NBP.

### 4.1.1 GPP and NPP

Global GPP and NPP simulated by ORCHIDEE-CNP averaged over the period 2001-2010 are 119 PgC
$yr^{-1}$ and 48 PgC $yr^{-1}$ respectively, both within ranges of the data-driven products listed in Table 1 (Appendix C; Table S2). GPP and NPP by ORCHIDEE-CNP are lower than in ORCHIDEE (140 Pg C $yr^{-1}$ for GPP and 60 Pg C $yr^{-1}$ for NPP). The values from ORCHIDEE are on the high end of the range of estimates from the data-driven products of Table 1.

Regarding the spatial distribution of GPP, ORCHIDEE-CNP simulated values lie for most parts of the
globe within the range given by the data-driven products (Fig. 2a). GPP simulated by ORCHIDEE-CNP is higher than the upper limit of data-driven estimates for some areas of Canada, Eastern USA, and Russia, and lower than the lower limit of data-driven estimates for some areas of South America and South Africa (Fig. 2a). The global spatial correlation coefficient ($R^2$) between modelled and data-driven GPP ranges between 0.59-0.60 for ORCHIDEE-CNP and between 0.87-0.91 for ORCHIDEE,
depending on the data-driven dataset considered (Fig. S1). The root-mean-square-error (RMSE) between modelled GPP and data-driven products is about twice higher in ORCHIDEE-CNP (437-498 g C $m^{-2}$ $yr^{-1}$) than in ORCHIDEE (250-324 g C $m^{-2}$ $yr^{-1}$). Compared to each of the three data-driven estimates of GPP, ORCHIDEE-CNP tends to overestimate low GPP values (<1500 g C $m^{-2}$ $yr^{-1}$) but underestimates high GPP ones (>2000 g C $m^{-2}$ $yr^{-1}$) (Fig. S1).

Regarding the spatial distribution of NPP, ORCHIDEE-CNP simulated values lie within the ranges of data-driven estimates for tropical regions and most of northern high-latitudes. However, NPP simulated by ORCHIDEE-CNP is higher than the upper limit of the range of data-driven estimates for temperate and western Europe (Fig. 2b). The spatial correlation between modelled NPP and satellite-based products is lower than for GPP, with $R^2$ ranging from 0.36 to 0.50 for ORCHIDEE-CNP and 0.71-0.79
for ORCHIDEE, depending on the data-driven datasets (Fig. S2). Unlike for GPP, the RMSE of NPP is substantially lower in ORCHIDEE-CNP (242-307 g C $m^{-2}$ $yr^{-1}$) than in ORCHIDEE (675-819 g C $m^{-2}$ $yr^{-1}$). Compared to each data-driven estimate of NPP separately, ORCHIDEE-CNP tends to overestimate low NPP values (<1000 g C $m^{-2}$ $yr^{-1}$) but underestimates high NPP (>1000 g C $m^{-2}$ $yr^{-1}$), a similar bias than noticed above for GPP (Fig. S2).

### 4.1.2 $CO_2$ fertilization effect for natural biomes

The main driver of the land carbon sink in global models is the increasing atmospheric $CO_2$ concentration (Kimball, 1983; Allen, 1994; Zhu et al., 2017). However, ecosystem-scale experiments indicate that nutrient constraints on plant functioning reduce the sensitivity of plant productivity to elevated $CO_2$ to such an extent that either the increase of productivity lasts only for few years (e.g.

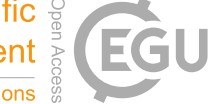

Norby et al., 2010) or no increase is observed (Ellsworth et al., 2017). We compare the simulated response of plant productivity to increasing $CO_2$ during the historical period (i.e., $CO_2$ fertilization effect $E_{co2}$) to observation-based estimates for C3 plants from historical change of deuterium isotopomers in leaf herbarium samples (Ehlers et al., 2015) and for global (C3 and C4) vegetation to indirect evidence from carbonyl sulfide (COS) atmospheric ice-core observations (Campbell et al.,

2017). The $CO_2$ fertilization effect is here defined by the GPP ratio ($E_{co2}$):

$$E_{co2} = \frac{GPP_{396}}{GPP_{296}},\qquad\qquad(3)$$

where $GPP_{296}$ indicates pre-industrial GPP (g C m$^{-2}$ yr$^{-1}$) under $CO_2$ concentration of 296 ppm and $GPP_{396}$ under current $CO_2$ concentration of 396 ppm. Those $CO_2$ concentrations of 296 ppm and 396ppm correspond to tropospheric mixing ratio of $CO_2$ in year ~1900 and 2013 respectively, similar to

values used for estimating the response of GPP to a ~100 ppm $CO_2$ increase in Ehlers et al. (2015) and Campbell et al. (2017).

Modeled $E_{co2}$ by ORCHIDEE-CNP of natural biomes ranges between 1.0 and 1.3 for most regions (Fig. 3a), slightly lower than global $E_{co2}$ derived from COS of 1.26-1.36 (Campbell et al., 2017). Modeled $E_{co2}$ for C3 plants (Fig. 3c, Fig. S3) are also consistent with $E_{co2}$ from herbarium samples (Ehlers et al.,

2015) equal to 1.23. When compared to ORCHIDEE without nutrient cycles, we found that ORCHIDEE-CNP simulates smaller and more realistic values of $E_{co2}$ (Fig. 3c, d), yet with lower values in boreal regions that could not be checked against observations (Fig. S3).

### 4.1.3 Inter-annual and seasonal variations of GPP

Inter-annual and seasonal variations of GPP reflect the response of ecosystems to inter-annual or

seasonal climatic variability, including the effect of natural (e.g. fires, wind throw, insect outbreaks, and storms) and anthropogenic disturbances (e.g. land management and land cover change) (Anav et al., 2015). N and P constraints on biological activity can modulate this response (Chen et al., 2011; Goll et al., 2018). Inter-annual anomalies of de-trended global GPP for natural and managed vegetation in the observation-driven models BESS-GPP and MTE-GPP for the period 2001-2011 range between -6 to 10

g C m$^{-2}$ yr$^{-1}$ and -17 to 22 g C m$^{-2}$yr$^{-1}$, respectively (Fig. 4a). Simulated anomalies in ORCHIDEE-CNP vary from -7.6 to 9.5 g C m$^{-2}$ yr$^{-1}$ and from -16 to 20 g C m$^{-2}$ yr$^{-1}$ in ORCHIDEE, thus being of lower magnitude when nutrients are included in the model. The modelled global GPP by ORCHIDEE-CNP shows a stronger inter-annual correlation than ORCHIDEE with BESS GPP over 2001-2011 ($R^2$=0.71, p=0.0011, RMSE=2.9 g C m$^{-2}$ yr$^{-1}$ against $R^2$=0.47, p=0.02, RMSE=7.7 g C m$^{-2}$ yr$^{-1}$). Both model

versions show weaker correlations with MTE-GPP (Fig. 4) which was noticed to underestimate the magnitude of GPP anomalies. Despite large discrepancies between the two observation-based estimates, the temporal correlations between each estimate and GPP from ORCHIDEE-CNP are higher than in ORCHIDEE for the northern Hemisphere (NH 30°N-90°N) but lower for other regions.

In contrast to the simulated inter-annual variations, the mean seasonal variations of GPP show a lower

agreement with BESS-GPP and MTE-GPP for ORCHIDEE-CNP than for ORCHIDEE. For the NH, ORCHIDEE-CNP simulates an earlier increase in GPP in spring and a delayed decline in GPP during





late summer and fall (July to November) compared with BESS-GPP, MTE-GPP and ORCHIDEE (Fig. 5a). The peak GPP in July is 22~36% lower in ORCHIDEE-CNP than in the data-driven products and ORCHIDEE. The mean variation of NH leaf area index (LAI, Table 1) for natural and managed vegetation in ORCHIDEE-CNP shows similar biases than for GPP (earlier increase, lower peak, delayed decline) when compared to GIMMS-LAI (Table 1 and Fig. S4). For tropical regions (30°N-30°S) and the southern hemisphere (30°S-90°S) ORCHIDEE-CNP simulates a smaller amplitude of the seasonal cycle compared to data-driven products and ORCHIDEE (Fig. 5b, c).

#### 4.1.4 Net biome productivity

Net biome productivity (NBP) is defined as the net C exchange between the atmosphere and the terrestrial biosphere, that is the sum of net primary productivity, heterotrophic respiration and emissions due to disturbances; positive values denoting a land carbon sink. ORCHIDEE-CNP simulates a much lower global NBP ($0.33 \pm 0.80$ Pg C yr$^{-1}$) over the period 1990-2016 than the JENA and CAMS atmospheric inversions (Table 1) giving a net sink of $1.44 \pm 0.93$ Pg C yr$^{-1}$ for JENA, $2.17 \pm 0.90$ Pg C yr$^{-1}$ for CAMS. NBP by CTracker atmospheric inversions (Table 1) is only available for period 2001-2016, but is close to JENA-inversion estimate during this period. Global simulated NBP is also smaller than the mean from 16 Trendyv6 land surface models involved in the Global Carbon Budget (Trendy v6) ($1.76 \pm 0.90$ Pg C yr$^{-1}$) (Le Quéré et al., 2018), but still falling within the 1-sigma standard deviation of NBP from those for 72% of the years and within the 2-sigma range for all years (Fig. 6a).

For the northern hemisphere (30°N-90 °N), NBP simulated by ORCHIDEE-CNP is a smaller sink -0.2 ~0.8 PgC yr$^{-1}$ (Fig. 6b) than in the JENA and CAMS-inversions for the period 1990-2016 (0.8~1.7 and 1.1~ 3.2 PgC yr$^{-1}$). The northern NBP is also a smaller sink than the mean of the Trendy v6 models (0.7~1.5 PgC yr$^{-1}$). For tropical regions (30°N~ 30°S), NBP simulated by ORCHIDEE-CNP is also lower than the mean of Trendy v6 models, but it is still a net sink for most of years, higher than in the inversions (Fig. 6c). For the southern hemisphere (30°S-90°S), NBP simulated by ORCHIDEE-CNP simulates a neutral C balance in line with estimates from inversions, while the Trendy v6 models simulate on average a small C sink (Fig. 6d). Therefore, the cause for the underestimation of the global C sink in ORCHIDEE-CNP during the last decades is the marginal NH C sink.

#### 4.2 Evaluation of N and P cycling

In the following, we evaluate the nutrient fluxes based on site measurements (Sullivan et al., 2014; Helfenstein et al., 2018) and data-driven products (Mayorga et al., 2010; Wang et al., 2018; Peng et al., 2019; Sun et al., 2020) (Table 1) for large spatial regions.

#### 4.2.1 Global nutrient and carbon cycling within natural ecosystems

We compared the simulated fluxes of N, P and C within ecosystems for the period 2001-2010 to the data-driven estimates from GOLUM-CNP (Table 1; Appendix B) on the global scale and for natural ecosystems at biome-scale. Modelled global C, N and P fluxes in ORCHIDEE-CNP are lower than the estimates by GOLUM-CNP (Fig. 7b, e), partly due to a smaller natural biome cover in ORCHIDEE-CNP. When these fluxes are compared on a per area basis, ORCHIDEE-CNP are consistent with





GOLUM-CNP values within their uncertainties (Fig. 7c, f). One exception is that ORCHIDEE-CNP
simulates a six-fold higher P leaching from soils (23 mg P m$^{-2}$yr$^{-1}$) than GOLUM-CNP (3.7 ± 9.7 mg P
m$^{-2}$yr$^{-1}$) (Fig. 7c, f).

**4.2.2 Openness of N and P cycles and nutrient residence time**

Nutrients taken up by plants are either recycled within the ecosystem or acquired from external sources
(P weathering of primary and secondary minerals, atmospheric N and P deposition, biological nitrogen
fixation (BNF), and N and P fertilizer addition to cultivated lands). Wang et al. (2018) calculated an
indicator of the openness of N and P cycling in natural ecosystems as the ratio of external inputs of N
and P into the ecosystem to the total amount of N and P that plants use for GPP. Similarly, we
diagnosed the openness for N and P ($O_N$ and $O_P$) from the ORCHIDEE-CNP output by:

$$O_x = \frac{I_x}{F_x + RSB_x} ,$$ (4)

where $I_x$ is the annual external nutrient input (gX m$^{-2}$ yr$^{-1}$), $F_x$ the annual plant uptake of soil nutrients
(gX m$^{-2}$ yr$^{-1}$), and $RSB_x$ the flux of nutrients recycled within plants (gX m$^{-2}$ yr$^{-1}$) by foliar nutrient
resorption prior to leaf shedding. External nutrient inputs include atmospheric N deposition and
biological N fixation (BNF), and include P deposition and P release from rock weathering.

Modelled $O_N$ in natural biomes by ORCHIDEE-CNP showed only a small variance across the globe,
whereas GOLUM-CNP predicts a higher $O_N$ in tropical and temperate regions than in boreal regions
(Fig. 8a, b). $O_P$ values are below 15% in ORCHIDEE-CNP for most biomes, of similar order of
magnitude than in GOLUM-CNP (Fig. 8c, d). ORCHIDEE-CNP simulates a lower $O_N$ in tropical
natural biomes than GOLUM-CNP, which is mainly due to lower but more realistic tropical BNF in
ORCHIDEE-CNP compared to GOLUM-CNP (see 4.2.3). ORCHIDEE-CNP simulates a higher $O_N$ in
high latitudes grassland (Fig. 8a, b) than GOLUM-CNP, which is due to overestimation of BNF in NH
in ORCHIDEE-CNP (see 5.1.3). Modelled $O_P$ in natural biomes by ORCHIDEE-CNP compares well
with GOLUM-CNP except for central Africa (Fig. 8c, d). This is primarily because ORCHIDEE-CNP
used a lower P deposition forcing than GOLUM-CNP.

Residence time quantifies the average time it takes for a N (or P) molecule from entering to leaving the
ecosystem ($\tau_N$ and $\tau_P$). In this study, we adopted the approach of Carvalhais et al. (2014) for the
carbon residence time. We define the residence time of N and P as the ratio of total respective nutrient
stock in the ecosystem to their respective total input flux:

$$\tau_N = \frac{\sum_{i=1}^{5} N_i + N_{inorg}}{N_d + BNF} ,$$ (5)

$$\tau_P = \frac{\sum_{i=1}^{5} P_i + P_{inorg}}{P_d + P_w} ,$$ (6)

where $N_i$ indicates the N mass (g N m$^{-2}$) in organic matter pools i (with i = plant, litter, SOM pools);
$N_{inorg}$ is the sum of all inorganic N pools, $N_d$ and $BNF$ are N deposition and biological N fixation rates





respectively (g N m$^{-2}$ yr$^{-1}$). Similarly, P$_i$ is the P mass (g P m$^{-2}$) in organic matter pools, $P_{inorg}$ the sum of inorganic P pools, and $P_d$ and $P_w$ are P deposition and P weathering release rates, respectively (g P m$^{-2}$ yr$^{-1}$).

Modeled median $\tau_N$ of natural biomes in ORCHIDEE-CNP varies between 56-1585 years, while $\tau_P$ varies within a large range of 101 to 223870 years (Fig. 9). ORCHIDEE-CNP captured the order of magnitude of $\tau_N$ and $\tau_P$ for forests found in GOLUM-CNP. Longer median $\tau_N$ (1585 years) and $\tau_P$ (1223870 years) are simulated for boreal forest compared to temperate and tropical forests (251-794 years for $\tau_N$ and 891-7080 years for $\tau_P$) and grassland (56-158 years for $\tau_N$ and 101-468 years for $\tau_P$)

by ORCHIDEE, consistent with results from GOLUM-CNP. However, for grasslands, simulated $\tau_N$ (56-158 years) and $\tau_P$ (101-468 years) are 5-11 folds shorter than in GOLUM-CNP (Fig. 9), mainly due to the smaller soil organic N stock and the lower external P inputs in ORCHIDEE-CNP (Fig. S12e, f, k, l).

**4.2.3 Biological nitrogen fixation**

Biological nitrogen fixation (BNF) is the major natural input of N to terrestrial ecosystems (Vitousek et al., 2013) and its temperature dependence is critical for large-scale patterns of N limitations (Du et al., 2020) while its response to changing conditions critically affects the occurrence of N limitation under increasing $CO_2$ (Goll et al., 2017b). Here we used BNF estimates from Sullivan et al. (2014) for tropical forests and a model-based global gridded BNP from Peng et al. (2019) to evaluate BNF in ORCHIDEE-

CNP. A special attention is given to tropical forests where interactions between N and P are potentially controlling BNF (Houlton et al., 2008).

Sullivan et al. (2014) estimated symbiotic and free-living BNF of tropical forest sites based on acetylene reduction assays with nodules, soil and litter at 12 sites, using a spatial sampling method to generate unbiased estimates of mean symbiotic BNF independent of legume trees abundance, which

more robustly captures the irregular distribution of nodules in the landscape. They found plot-level measurements of BNF rates which were five times lower than used in previous empirical upscaling to the tropical forest biomes (Cleveland et al. 1999; Wang and Houlton, 2009). Recalibrating the empirical relationship between BNF and either NPP or evapotranspiration from Cleveland et al. (1999) and Wang and Houlton et al. (2009) based on their plot level estimates of BNF, Sullivan et al. (2014) upscaled

plot-level BNF to all tropical forests to 4.9~6.3 kg N ha$^{-1}$ yr$^{-1}$ and 3.7~7.8 kg N ha$^{-1}$ yr$^{-1}$ for first and third quantiles, respectively. Peng et al. (2019) developed a global BNF model constrained by observed C:N ratios of various plant pools (Appendix D1), and they simulated a higher BNF for tropical forests than Sullivan et al. (2014) with an interquartile range (IQR) of 14~43 kg N ha$^{-1}$ yr$^{-1}$. BNF by ORCHIDEE-CNP in tropical forests is comparable in median value (5.6 kgN ha$^{-1}$ yr$^{-1}$) but has a wider

IQR of 2.5-7.8 kg N ha$^{-1}$ yr$^{-1}$ than the upscaled estimation according to Sullivan et al. (2014) (Fig. 10), but is lower than Peng et al. (2019)'s estimate, even lower than their first quartile. For all biomes, ORCHIDEE-CNP simulated a higher total BNF (153 Tg N yr$^{-1}$) than Peng et al. (2019) (116 Tg N yr$^{-1}$) and a roughly comparable spatial range (IQR = 3.1-18 kg N ha$^{-1}$ yr$^{-1}$ compared to 0.23-15 kg N ha$^{-1}$ yr$^{-1}$) (Fig. 10).





### 4.2.4 N and P load rates to rivers on catchment scale

ORCHIDEE-CNP simulated a global $N_{load}$ calculated over the period 2002-2010 of 55 Tg N yr$^{-1}$ with $N_{load}$ from natural biomes being 35 Tg N yr$^{-1}$, close to the estimate by GOLUM-CNP (38 Tg N yr$^{-1}$) but higher than one by GlobalNEWS2 (28 Tg N yr$^{-1}$ in year 2000). Simulated $N_{load}$ from managed land, namely croplands and pastures, is 21 Tg N yr$^{-1}$, about half the estimate from GlobalNEWS2 (40 Tg N yr$^{-1}$ in year 2000; Mayorga et al., 2010). On catchment scale, $N_{load}$ in the year 2000 simulated by ORCHIDEE-CNP have a comparable range (0~2500 kg N km$^{-2}$ yr$^{-1}$) but generally lower values than GlobalNEWS2 (Fig. 11a to c), excepted for some basins dominated by Oxisols such as the Amazon (Fig. S12). The higher Amazon $N_{load}$ in ORCHIDEE-CNP is likely explained by the omission of the high sorption capacity for N of tropical soils as ORCHIDEE-CNP assumes globally uniform sorption capacities.

Global total $P^*_{load}$ ($P^*_{load} = P_{dra} + P_{runoff}$; section 3.2.3) is estimated at 3.8 Tg P yr$^{-1}$ averaged over the period 2002-2010 in ORCHIDEE-CNP. Natural biomes account for only ~10% of global total $P^*_{load}$ (0.39 Tg P yr$^{-1}$), about an order of magnitude less than in GOLUM-CNP (Table S3). Croplands and pastures account for most of the global total $P^*_{load}$ with a value of 3.3 Tg P yr$^{-1}$ which is rather close to the estimate of 4 Tg P yr$^{-1}$ by Bouwman et al. (2013b), but lower than the 5.4 Tg P yr$^{-1}$ reported by Lun et al. (2018) and higher than in Mekonnen and Hoekstra (2018) (0.6 Tg P yr$^{-1}$) (Table S5). The simulated values of $P_{load}$ (Eq. (2)) at catchment scale are in the same order of magnitude than estimates from GlobalNEWS2 (Mayorga et al., 2010). However, ORCHIDEE-CNP simulates generally lower $P_{load}$ over the Amazon and Central African basins, and higher $P_{load}$ in European catchments with intensive fertilizer use (> 0.5g N yr$^{-1}$ m$^{-2}$) compared to GlobalNEWS2 (Mayorga et al., 2010) (Fig. 11d, e).

### 4.2.5 Soil solution inorganic phosphorus turnover rate

The turnover of P in the soil solution is an important component of P bioavailability in soil (Helfenstein et al., 2018). We compared soil solution P turnover in ORCHIDEE-CNP with empirical data from isotope exchange kinetic (IEK) experiments (Fardeau et al. 1991; Frossard et al. 2011). IEK experiments measure the exchange rate of inorganic P between soil solution and the soil solid phase ($K_m$, unit: min$^{-1}$) in laboratory conditions, omitting biological processes. The inorganic exchange processes captured in $K_m$ dominate short-term P fluxes.

For a straightforward comparison (excluding biological uptake process from model output to match the experimental method used for measuring), we diagnosed $K_m$ (unit: min$^{-1}$) in the simulations from the net exchange between dissolved and sorbed labile P:

$$K_m = \frac{P_{dissolved}}{\Delta P_{sorbed} - F_{dss}}, \tag{7}$$

Where $P_{dissolved}$ is the labile P in soil solution (g P m$^{-2}$) and $\Delta P_{sorbed}$ the change in labile P adsorbed (g P m$^{-2}$ t$^{-1}$) with t=30 minutes (model time step) corrected for the loss term of $P_{sorbed}$ related to strongly sorption. In ORCHIDEE-CNP, the exchange between $P_{dissolved}$ and $P_{sorbed}$ ($F_{dss}$) is calculated using a

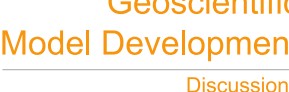
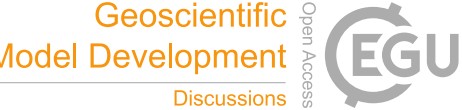

Freundlich adsorption isotherm by assuming a chemical equilibrium between both pools is reached on the model time step (Goll et al., 2018).

We used data from a global compilation of empirical $K_m$ data from 218 different soil samples, mainly from Europe and North America but also covering Asian and Africa (Helfenstein et al., 2018). We
found that simulated $K_m$ are of similar magnitude with median value of 24~ 468 min$^{-1}$ across soil orders compared to those empirical data with simulated 5~990 min$^{-1}$ (Figure 12). However, the turnover rates for different soil types did not match empirical data. The model overestimated $K_m$ for Alfisols, Aridisols, Entisols, Mollisols, Spodosols, Ultisols, and underestimated $K_m$ for Oxisols and Vertisols.

### 4.2.6 P biochemical mineralization from phosphatase enzymes

Biochemical phosphorus mineralization (BPM) mediated by extracellular phosphatase enzymes produced plants and microbes through cleaving P out of organic matter, is a potentially significant pathway for mineralizing P (Wang et al., 2007). ORCHIDEE-CNP includes an empirical parameterization of BPM which accounted for an enhanced mineralization when plants experience suboptimal P-to-N availabilities, as an approximation of the stoichiometric status of the whole
ecosystem, including the effect of substrate availability on mineralization (McGill and Cole, 1981) (see Eq. 18 in Goll et al., 2017b). Potential phosphatase activity ($P_{ases}$) can be measured in the lab from soil samples, and serves as a surrogate for the lack of direct BPM measurements in real-world soil conditions (Sun et al., 2020). We compare the global pattern of simulated P mineralization fluxes in ORCHIDEE-CNP with the map of $P_{ases}$ extrapolated from site measurements (Margalef et al., 2017)
produced by Sun et al. (2020). The modelled BPM is higher in tropical regions than in desert and boreal regions, consistent with the $P_{ases}$ pattern (Fig. 13), although the scarcity of tropical measurements makes the high values observed in these regions more uncertain (Sun et al., 2020).

### 4.2.7 Biome-scale nutrient budgets

In a model resolving nutrients like ORCHIDEE-CNP, as changes in climate and $CO_2$ drive an increased
land carbon storage, N and P tend to accumulate over time in biomass and soil carbon pools. We analyzed the modelled N and P storage changes for 4 managed and 10 natural biomes for the period of the 2000s from the ORCHIDEE-CNP output. We found a net N accumulation of 51.5 Tg N yr$^{-1}$ globally in natural biomes mainly fueled by BNF (Fig. 14a), except for boreal forests and natural grasslands where the contribution of N deposition is the dominant source of new N inputs. A net N accumulation
of 14.4 Tg N yr$^{-1}$ is found in pastures, whereas a net N loss of 6.5 Tg N yr$^{-1}$ is found in croplands (Fig. 14a). N fertilizers application accounts for 20~33% of the total pasture N input, and dominates N inputs in cropland (80~95%, 149 Tg N yr$^{-1}$ in total). Only 55~60% of N input from fertilizers application is harvested in crop biomass. The rest of these inputs being lost in gaseous form (45.7 Tg N yr$^{-1}$) and from drainage and surface runoff leaching (20.7 Tg N yr$^{-1}$).

We found a net P accumulation of 0.8 Tg P yr$^{-1}$ in tropical and temperate forests, but a net P loss of -0.16 Tg P yr$^{-1}$ in natural grasslands due to increased soil P fixation exceeding P deposition and weathering inputs. Boreal forests show a nearly balanced P budget (Fig. 14b). For tropical broad-leaved



evergreen forests, inputs from deposition and weathering of primary and secondary minerals represent 55% and 45% of total P inputs, respectively. For other natural biomes, deposition dominates (>75%) the
P inputs. P losses to soil P fixation are the major P loss pathway for all biomes (Fig. 14b).

A net P accumulation of 1.2 Tg P yr$^{-1}$ is simulated for fertilized pasture and croplands, respectively (Fig. 14b). Manure fertilization dominates the total pasture P input (69%~93%) but mineral fertilizers dominates the total cropland P inputs (>95%). In croplands, 31% of cropland P input from fertilizers is harvested in crop biomass (5.9 Tg P yr$^{-1}$), 36% is lost to soil P fixation (4.3 Tg P yr$^{-1}$) and drainage and
surface runoff (2.6 Tg P yr$^{-1}$). The net P accumulation in pasture is mainly attributed to the omission of P output by grazing in ORCHIDEE-CNP.

### 4.3 Evaluation of leaf and soil stoichiometry

The N:P ratio of leaves and soil organic matter is an indicator of the relative availabilities of N and P, as direct measurement of availabilities are lacking, providing important information about nutrient
constraints on ecosystem functioning (Sterner and Elser, 2002; Ågren, 2008). In this section, we evaluate modelled leaf and soil N:P ratio at global and biome scale.

#### 4.3.1 Leaf N:P ratios for natural biomes

Leaf N:P ratios for natural biomes predicted by ORCHIDEE-CNP vary between 15~25 (Fig. 15a). The observed decline in median leaf N:P ratios with increasing latitude was not reproduced by the model
(Appendix E1; Fig. 15e), although the modelled latitudinal distribution of leaf N:P ratios remained within the 10~90th quantiles of the site level data (Kerkhoff et al., 2005; McGroddy et al., 2004; Reich and Oleksyn, 2004). Further, the simulated leaf N:P ratios fall within the interquartile of upscaled site measurements by Butler et al. (2017) for most of the globe, with the exception of regions north of 55$^{\circ}$N where leaf N:P are outside the observation-based range, suggesting a too strong P constraint relative to
N (Fig. 15a, e).

#### 4.3.2 Soil C:N, C:P and N:P ratios for natural biomes

We evaluate here the modelled C:N, C:P and N:P ratios of soil organic matter for different biomes against data from the large compilation of measurements for soils (0-60cm depth) by Tipping et al. (2016). Modelled C:N ratios fall into much more narrow ranges (7.8~11.8 for the widest IQR)
compared to the observations (11.1~20.5, Fig. 16a), as a result of prescribing constant C:N ratios in ORCHIDEE-CNP (Goll et al., 2017a). SOM P content varies in ORCHIDEE-CNP as a consequence of varying BPM rates and thus C:P and N:P ratios of SOM show pronounced variation in space. ORCHIDEE-CNP simulates comparable N:P ratios than measurements in terms of both median value and distributions for tropical forests, but overestimates the observed N:P ratios by 108-327% in
temperate forests, tropical and temperate grasslands soils (Fig. 16b, c). The higher observed C:P and N:P in forest compared to grassland soils are not captured by ORCHIDEE-CNP (Fig. 16b, c). We also compared ORCHIDEE-CNP N:P ratios to the results of GOLUM-CNP which were based on the data





from Zechmeister-Boltenstern et al. (2015), more limited than Tipping et al. (2016) and found an overestimation for temperate forests, tropical and temperate grasslands.

**4.4 Evaluation of resource use efficiencies**

We evaluate here the resource use efficiencies of GPP for light (L), water (W), C, N and P defined by:

$$LUE = \frac{GPP}{fAPAR \times PAR},$$ (8)

$$WUE = \frac{GPP}{ET},$$ (9)

$$CUE = \frac{NPP}{GPP},$$ (10)

$$NUE = \frac{GPP}{F_N},$$ (11)

$$PUE = \frac{GPP}{F_P},$$ (12)

Where GPP is the annual gross primary productivity (g C m$^{-2}$ yr$^{-1}$), fAPAR the fraction of absorbed photosynthetically active radiation (%), PAR is annual Photosynthetically Active Radiation (W m$^{-2}$ yr$^{-1}$), ET the annual evapotranspiration (mm m$^{-2}$ yr$^{-1}$), $F_N$ and $F_P$ the total N uptake (g N m$^{-2}$ yr$^{-1}$) and P uptake by plants (g P m$^{-2}$ yr$^{-1}$), respectively. We calculated fAPAR in ORCHIDEE-CNP and ORCHIDEE as a function of leaf area index (LAI): fAPAR=1-exp(-0.5·LAI) (Ito et al., 2004).

Compared to observed LUE (section 3.2.2; Appendix D; Jung et al., 2009; Baret et al., 2013), ORCHIDEE-CNP modelled median values at biome level are generally lower, but still within the ranges of uncertainties of observation-based datasets excepted for tropical (TRF) and temperate deciduous forests (TEDF). We found that ORCHIDEE-CNP underestimates LUE in TRF and has higher values in boreal forests (BOCF). Overall, ORCHIDEE-CNP simulated LUE values which are comparable to observation-based datasets, but it has a lower agreement than ORCHIDEE for the differences among biomes (Fig. 17a).

Compared to observed WUE (section 3.2.3), ORCHIDEE and ORCHIDEE-CNP simulated values fall within the uncertainty range of observations. However, the WUE values from ORCHIDEE-CNP are on the high end of the range for temperate conifers (TECF) and BOCF and on the low end for temperate and tropical grasslands (TEG and TRG). The highest median WUE were correctly simulated in temperate forests by ORCHIDEE-CNP (Fig. 17b), but the lowest WUE values were simulated in temperate instead of tropical forests.

Compared with observed CUE (section 3.2.3), ORCHIDEE-CNP simulated comparable values for TEDF and TECF but lower values for TRF, BOCF and grasslands. Both ORCHIDEE-CNP and ORCHIDEE cannot capture the increase of CUE from tropical to boreal forests apparent in the observation-based products (Fig. 17c) and in measurements from forest sites (Piao et al., 2010).



Although ORCHIDEE-CNP simulates a larger spread in CUE than ORCHIDEE, the spread is more
narrow than based on MODIS products.

Consistent with site-observations of NUE from Gill and Finzi (2016) and GOLUM-CNP outputs,
ORCHIDEE-CNP simulated correctly the high values of TECF and the low values of tropical forests.
However, compared with site-observations of PUE from Gill and Finzi (2016) showing a PUE decrease
from tropical to boreal region, ORCHIDEE-CNP simulated a rather flat value. This suggests a too
strong P limitation in high latitude ecosystems, consistent with the fact that the model underestimates
peak northern GPP (section 4.1.3) and the northern land sink (section 4.1.4). Nevertheless, the model
simulated PUE values falling in the range of GOLUM-CNP estimates. Tropical C4 grasslands have
higher simulated NUE and PUE than temperate C3 grasslands, consistent with GOLUM-CNP (Fig. 18).

## 5. Discussion

By performing a detailed evaluation of ORCHIDEE-CNP against carbon observations, we found that
including nutrients in this LSM overall did not degrade the performances of the model in general.
Specifically, ORCHIDEE-CNP simulates a more realistic sensitivity of plant productivity to increasing
$CO_2$ and inter-annual variation of GPP in the northern hemisphere (section 4.1.2). However,
ORCHIDEE-CNP shows lower performances than ORCHIDEE for the seasonal cycle of GPP in
tropical regions and underestimates the peak GPP in the NH growing season as well as the northern land
sink during recent decades (section 4.1.3 and 4.1.4). The underlying cause of the degradation of the
model performance for NH carbon cycling is most likely an underestimation of plant P availability as
indicated by the bias in leaf stoichiometry and resource use efficiencies as a consequence of a
pronounced P immobilization in accumulating soil organic matter. In section 5.1 we discuss how N and
P affect C cycling and identify key processes related to model biases in C cycling. Despite the
mentioned biases in P plant availability, ORCHIDEE-CNP generally simulates global total N and P
fluxes (e.g. BNF, P weathering, N and P uptake etc.) comparable to the GOLUM-CNP fusion of
observation-based datasets and to individual datasets. Processes related to biases in nutrient fluxes are
discussed in section 5.2-5.4.

### 5.1 Terrestrial C cycle

#### 5.1.1 N and P constraints on $CO_2$ fertilization effect

The effect of $CO_2$ fertilization on terrestrial ecosystems productivity is thought to be the dominant
driver behind the current land carbon sink. Although all models include a $CO_2$ fertilization effect on
GPP, the strength differs strongly between models (Friedlingstein et al., 2014). Further, how a $CO_2$
fertilization effect on GPP translates into an increase of land carbon storage depends not only on the
response of GPP but also on the residence time of carbon in ecosystems, leading to model differences in
the contribution of increasing $CO_2$ to the land carbon sink in the past decades (Huntzinger et al., 2017;
Liu et. al., 2019). We used proxies of the historical increase in GPP for an indirect model evaluation of
the $CO_2$ fertilization effect from COS and deuterium measurements of herbarium samples (Ehlers et al.,
2015; Campbell et al., 2017), and found that ORCHIDEE-CNP has smaller and more realistic $E_{co_2}$ than





the same model without nutrients (Fig. 3), in particular for C3 plants and in boreal regions (Fig. S3). Both ORCHIDEE-CNP and ORCHIDEE simulated a $Eco_2$ for C4 grass of ~1, as the carboxylation of C4 plants is weakly influenced by elevated $CO_2$ (Osmond et al. 1982; Pearcy and Ehleringer, 1984; Bowes, 1993). Yet, despite a less realistic $Eco_2$, ORCHIDEE performs better than ORCHIDEE-CNP in simulating the net land sink and its trend. This result suggests that there are bias compensation effects in ORCHIDEE, for instance an underestimated residence time of carbon in ecosystems compensating for a too large $Eco_2$. With better observational constraints on both $Eco_2$ and carbon residence time (e.g. Walker et al., 2019), and more versions of models including nutrients, it should be possible in the future to examine more in detail such bias compensation effects across different models.

### 5.1.2 N and P constraints on inter-annual and seasonal variability of GPP

To what extent nutrient effects on vegetation affect the sensitivity of ecosystem $CO_2$ fluxes to climatic variation is unclear (Goll et al., 2018). For instance, drought can reduce nutrient use by decreasing GPP, but it also slows down decomposition which supplies nutrients for plant uptake. Further, N:P stoichiometry is also strongly modified by drought and warming towards increased N:P in whole plant biomass (Yuan and Chen, 2015). Here we found that the inclusion of N and P nutrient cycles in ORCHIDEE affects the inter-annual variability of GPP for all vegetation types. In ORCHIDEE-CNP, the inter-annual variation (IAV) of GPP is better correlated to that of observation-based datasets than in ORCHIDEE globally and for the NH, but less correlated for other regions (Fig. 4). Observation-based GPP estimates are uncertain, as some of them ignore soil moisture induced reductions of GPP during drought (Stocker et al., 2019), and soil thaw and snow-related effects (Jiang and Ryu, 2016). Thus, at the moment, it is difficult to falsify one model version over another, and to constrain nutrient effects on the variation of GPP, based on current observation-based GPP.

In order to further explore the underlying reasons of the general improvement in the IAV of GPP due to the inclusion of nutrient cycles, we analyzed the sensitivity of GPP anomalies to anomalies of temperature ($S_T$), precipitation ($S_P$) and shortwave radiation ($S_R$), all with mean annual values (Appendix I). We found that $S_P$ by ORCHIDEE-CNP compares well with BESS-GPP and MTE-GPP, while it is overestimated in ORCHIDEE (Figs. S5 and S6). Thus, the difference in $S_P$ is likely the major reason for the differences in IAV in NH between model versions, as $S_T$ and $S_R$ show only minor differences there. This provides confidence that the improvement of IAV of GPP in the NH is due an improved sensitivity towards a climatic driver (i.e. $S_P$). For tropical regions, ORCHIDEE-CNP simulates more realistic $S_P$ but higher biases in $S_R$ than in ORCHIDEE, while observation based estimates of $S_T$ disagree on the sign and model versions show only minor differences (Fig. S6). Therefore, the deterioration of the IAV of tropical GPP by the inclusion of nutrient cycles is likely caused by enhanced biases in $S_R$ due to a lowering of LUE of GPP (section 4.4 and 5.5).

The seasonal magnitude of GPP by ORCHIDEE-CNP for extratropical regions is smaller during the peak growing season (mid-May – mid-August for NH, mid-August – April for SH) and larger at the end of the growing season (after mid-August for NH, April – mid-April for SH) compared to BESS-GPP and MTE-GPP (Fig. 5a), whereas ORCHIDEE does not exhibit such a bias. The ORCHIDEE-CNP bias of GPP seasonality is mainly explained by a bias of LAI (Fig. S4a). For NH, the delayed increase in





LAI could be partly caused by nutrient shortage during the first half of the growing season, as indicated by the increasing modeled leaf nutrient concentration throughout the growing season (Fig. S4). Several factors could lead to a too low supply of nutrients in the beginning of the growing season: an insufficient internal plant nutrient reserve due to a too low resorption of nutrients prior to leaf shedding or an underestimation of nutrient uptake during the dormant season, an insufficient investment into root

growth to acquire nutrients, and an overestimation of soil nutrient losses during dormant season leaving the soil nutrient depleted at the beginning of the growing season. Many of the related processes (e.g. root phenology, mineralization, nutrient resorption, growth allocation) are only rudimentary represented. For tropical regions, we showed that ORCHIDEE-CNP simulates a quasi-flat seasonal cycle of GPP, in contrast with a peak of GPP during the wet season in MTE-GPP and BESS-GPP, which is correctly

captured by ORCHIDEE (Fig. 5b, c). The reduction of seasonal GPP in ORCHIDEE-CNP compared to ORCHIDEE is more pronounced in the dry season (~100 g C m$^{-2}$) than in the wet season (Fig. 5b, c), concurrent with a larger reduction of LAI in the dry season (Fig. S4). Tropical phenology is currently only rudimentary represented in ORCHIDEE(-CNP) (Chen et al., 2020) causing a suboptimal allocation of nutrients to leaves which could cause the biases in the seasonal cycle of GPP and LAI. Model-data

assimilation of phenology (Williams et al., 2009; MacBean et al., 2018; Bacour et al., 2019) and efforts to better characterize processes related to plant resource investment into different tissues and symbioses (Prentice et al., 2015; Warren et al., 2015; Jiang et al., 2019) and leaf age effects during the year for evergreen forests (Chen et al., 2020) should help to reduce tropical phenology biases in future versions of ORCHIDEE-CNP.

**5.1.3 N and P constraints on land carbon storage**

Current LSM unanimously conclude that $CO_2$ fertilization is the main driver of the land C sink and its trend (Friedlingstein et al., 2014), but it remains unclear to what extent other drivers (i.e. climate change, land management, nutrient deposition) contribute to the sink as well. Also, it remains unclear how commonly omitted dynamics (climate and management induced tree mortality, nutrients) lead to

overestimation of the contribution of $CO_2$ fertilization in models (Ellsworth et al., 2017; Fleischer et al., 2019). ORCHIDEE-CNP simulates a land carbon sink over the past decades that is lower than other DGVM models and atmospheric inversions. In particular, the NH carbon sink which persistently increased over the last 50 years (Ciais et al., 2019) is strongly underestimated, despite the fact that the response of GPP to $CO_2$ is in line with proxy data (see 5.1.1). The few Free Air Carbon Enrichment

(FACE) studies that have experimentally applied elevated $CO_2$ levels in mature stands found no increase in biomass production (Bader et al., 2013; Klein et al., 2016; Körner et al., 2005; Sigurdsson et al., 2013; Ellsworth et al., 2017), thus an increase in GPP does not necessarily translate into an increase in biomass production, whereas in most DGVMs where mortality is constant and growth follows GPP, biomass production is inevitably coupled to GPP. Based on upscaling of data from FACE experiments,

Terrer et al. (2019) suggested that the effect of elevated $CO_2$ on biomass may be severely overestimated (on average by a factor of 3.6) in land surface models which ignore nutrients. It would be tempting to conclude from this study that ORCHIDEE-CNP is 'right' in its underestimation of the carbon sink whereas other models are 'wrong' because they miss processes such as forest regrowth (Pugh et al., 2019) from e.g. decreased harvesting pressure (Ciais et al., 2008) and thus have a realistic NH land sink





for the wrong reasons. We also showed that ORCHIDEE-CNP underestimates peak GPP (section 4.1.3 and 5.1.2) and overestimates P limitations in the NH (section 4.3 and 5.4) thus, another explanation is that the NH sink in this study is too low because of too strong P limitations in this region. These two hypotheses explaining why we underestimate the NH sink (missing forest regrowth vs. too strong nutrient limitations in the NH) are examined below.

The too small NH carbon sink in ORCHIDEE-CNP may be explained by a too strong immobilization of nutrients in accumulating nutrient-rich organic matter, which leads to a reduction of plant available nutrients, the so-called 'progressive nutrient limitation' proposed by Luo et al. (2004) and subsequently to a reduced biomass production. The amount of accumulated N and P immobilized into SOM in the NH during 1850-2016 reaches up to 75.3 g N m$^{-2}$ and 2.4 g P m$^{-2}$ respectively, which is twice as much

as the accumulated respective nutrient inputs to ecosystems in this region during the same period (37.8 g N m$^{-2}$ and 1.6 g P m$^{-2}$; Figs. S8 and S9). This suggests a strong progressive nutrient limitation in the model. The omission of nutrient controls on litter and SOM decomposition in the soil module of ORCHIDEE-CNP could have favored the immobilization of nutrients in accumulating SOM (Zhang et al., 2018). Microbe incubation and N fertilization experiments showed that a low availability of

nutrients can hamper the built-up of SOM as more carbon gets respired by decomposers due to an elevated energetic requirements of processing low quality substrate (Recous et al., 1995; Janssens et al., 2010; Allison et al., 2009) and an overall lower microbial activity (Wang et al., 2011; Knorr et al., 2005). Uncertainties with respect to the capability of ecosystems to up-regulate P mineralization when P becomes scarce could have contributed to the decline in plant available nutrients with increasing SOM

stocks. The inclusion of nutrient effects on decomposition and microbial dynamics in ORCHIDEE-CNP is ongoing (Zhang et al., 2018, 2020) but the lack of a quantification of the ability of ecosystems to enhance P recycling hampers model developments.

The too small NH carbon sink in ORCHIDEE-CNP may also be explained by the lack of representation of effects of forest age and management on biomass turnover and biomass production efficiency (i.e.

CUE). Pugh et al. (2019) found that old-growth forests in the NH have a much smaller C sink than re-growing forests (<0.1 Pg C yr$^{-1}$ compared to 0.86 Pg C yr$^{-1}$) for the period 2001-2010. Forest management effects on biomass production efficiency and biomass turnover is only rudimentary represented in ORCHIDEE(-CNP). ORCHIDEE-CNP prescribes constant tree mortality rates (i.e. the fraction of total carbon in wood lost to litter) whereas in reality tree mortality rates change with

management and climate conditions (Peng et al., 2011). Moreover, ORCHIDEE(-CNP) omits the effect of forest age on C uptake. Compared to data-driven estimates for C storage (Appendix G and H), ORCHIDEE-CNP simulates a higher global above ground forest biomass (387 Pg C; 283 Pg C for GlobBiomass and 221 Pg C for GEOCARBON; Figure S14) but lower global soil organic carbon (801 Pg C; 4387 Pg C for Soilgrids and 1680 Pg C for GSDE; Figure S15).

**5.2 N and P budgets of natural and agricultural ecosystems**

In the following, we look into the state of the N and P cycle to assess if the underlying factors contributing to nutrient constraints on the carbon cycle discussed in 5.1 are plausible. We first discuss the in- and out- fluxes that control the nutrient capital of ecosystems, and second the efficiency at which



nutrients are recycled within ecosystems. Both factors control the availability of nutrients, and thus the
nutrients constraints on the C cycle.

### 5.2.1 Evaluation of in and out fluxes of N and P

The main N input to natural ecosystems are atmospheric N deposition, which is part of the model
forcing, and biological N fixation (BNF). ORCHIDEE-CNP simulates a tropical BNF which is close to
estimates derived from measurements, but a higher global BNF of 153 Tg N yr$^{-1}$ than previous estimates
(40-128 Tg N yr$^{-1}$) (Galloway et al., 2004; Vitousek et al., 2013), primarily due to an overestimation of
extratropical BNF rates (Fig. 10). The latter is likely related to the omission of a direct temperature
control on BNF in ORCHIDEE-CNP which was shown to be able to explain low BNF rates at higher
latitudes (Houlton et al., 2008).

ORCHIDEE-CNP accounts for two major loss pathways of N from the soil-plant system: gaseous N
emissions ($NH_3$, $N_2O$, $N_2$ and NO) and dissolved inorganic N exports through surface runoff and
drainage ($N_{load}$), but N losses via erosion and fire are ignored. We showed ORCHIDEE-CNP simulates
global $N_{load}$ from natural systems which are comparable with observed estimates (section 4.2.4).
However, ORCHIDEE-CNP simulates much lower global $N_{load}$ from cultivated land than
GlobalNEWS2. This underestimation is likely due to the lower manure N fertilizer scenario, as part of
the model forcing, than used in GlobalNEWS2 (Fig. S11), due to the omission of point sources such as
livestock farms and sewage systems. Thus, ORCHIDEE-CNP also simulates generally lower $N_{load}$ on
catchment scale than GlobalNEWS2 model, except for some tropical basins (which are less subject to
anthropogenic N inputs than higher latitudes) (Fig. 11a to c). The overestimation of $N_{load}$ in some
tropical basins dominated by Oxisols soils (e.g. Amazon) (Fig. S10) is mainly attributed to our
assumptions regarding the sum of $N_{runoff}$ and $N_{dra}$ to approximate $N_{load}$ (Eq. (1); see 3.2.2). Tropical
basins have strong N denitrification losses from the groundwater as a result of large flux and high N
concentration of groundwater (Bouwman et al., 2013a), thereby a part of the N is lost before the water
enters the river (i.e. $N_{dra}$). Denitrification losses from groundwater are not accounted for in
ORCHIDEE-CNP. In addition, ORCHIDEE-CNP applies a globally uniform ammonium sorption
capacity of soils, calibrated to a limited number of soil measurements (Zaehle and Friend, 2010) which
might not be applicable to highly weathered and low pH soils in the tropics.

Gaseous N emissions occur from N nitrification and denitrification processes within soils. We showed
that ORCHIDEE-CNP simulates comparable global gaseous N emission for natural biomes (90 Tg N yr$^{-1}$)
than GOLUM-CNP (67 Tg N yr$^{-1}$) (Fig. 7). The modeled range of gaseous N emission by
ORCHIDEE-CNP is consistent with GOLUM-CNP for TRF, TRG and TEG, but too high for TEDF,
TECF and BOCF (Fig. S12). This can be mainly attributed to the higher N mineralization rates in
ORCHIDEE-CNP, related to the higher substrate availability, (i.e. N in litter and SOM) (Fig. S12).

Global total $P^*_{load}$ from natural lands in ORCHIDEE-CNP is lower than estimates by GOLUM-CNP
(Table S3). This lower total $P^*_{load}$ for natural biomes mainly occurs in forest ecosystems (Fig. S12)
which can be mainly attributed to the low substrate availability (i.e. dissolved P concentration) due to
strong P immobilization into biomass and SOM for forests (Fig. S12a, g; see section 5.1.3) in





ORCHIDEE-CNP. Global total $P^*_{load}$ from managed lands in ORCHIDEE-CNP (3.8 Tg N yr⁻¹) is lower than estimates based on mass-balance (Bouwman et al., 2013b; Lun et al., 2018) (4-5.4 Tg P yr⁻¹), which is expected as ORCHIDEE-CNP cannot capture the high P losses from extremely high livestock
densities in small areas where soil P retention capacities have been reached (see below). Besides, Lun et al. (2018) considered sludge inputs (1.4 Tg P yr⁻¹ in year 2000), which are omitted in ORCHIDEE-CNP. ORCHIDEE-CNP simulates generally lower $P_{load}$, in particular in Amazon and Central Africa catchments compared with GlobalNEWS2. This underestimation can be partly attributed to the strong P fixation capacity of soil (i.e. occlusion) in tropical regions in ORCHIDEE-CNP, which is related to a
fast turnover of soil inorganic P (see 4.2.5 and 5.2.2). Note that $P_{dra}$ is small but non-negligible in surface and shallow subsurface runoff (Mayorga et al., 2010), and ignoring this flux for $P_{load}$ (section 3.2.2) may lead to our underestimation of $P_{load}$ (Eq. (2)). Simulated $P_{load}$ is also higher than GlobalNEWS2 in European catchments, where pasture is widespread (Fig. 11d, e), which can be attributed to either a less effective soil P sorption capacity or the omission of grazing which enhances
the mineralization of organic P and as a consequence inorganic P losses.

### 5.2.2. Evaluation of P mineralization and inorganic P dynamics

The simulated soil P fluxes cannot be evaluated against direct observations (Frossard et al., 2011). Here we used pool turnover times instead of fluxes to evaluate P dynamics. ORCHIDEE-CNP has similar orders of magnitude of inorganic P turnover rates in the soil solution ($K_m$), but cannot capture variations
in $K_m$ among different soil orders as derived from measurements (Helfenstein et al., 2018), due to that ORCHIDEE-CNP distinguishes only between three classes of soils with respect to parameter controlling inorganic P sorption dynamics: Oxisols, Molisols and other soils (Table S4; Goll et al., 2017a). The difference in biases of $K_m$ between Oxisols and other soil suborders suggests that our parameterization of sorption is the primary cause for the model bias, as all other processes affecting
dissolved P are not a function of soil type. Simplistic representation of inorganic P processes operating on longer timescale (occlusion, strong sorption) used in the few land surface model simulating the P cycle (Wang et al., 2010; Yang et al., 2014; Goll et al., 2017b) are primarily based on calibration rather than data driven, and remain a large source of uncertainty regarding changes in P availability under elevated $CO_2$ (Goll et al., 2012). Recent advances in the quantification of inorganic soil P turnover
(Helfenstein et al., 2020) should be used to improve this part of models.

Global organic P mineralization fluxes were evaluated against potential phosphatase activity ($P_{ases}$) data (Margalef et al., 2017; Sun et al., 2020), and actual bulk P mineralization measurements based on radioisotopic approaches (Bünemann, 2015; Wanek et al., 2019), to our knowledge the first attempt to evaluate a model for this process. We found that ORCHIDEE-CNP captured the general global pattern
of $P_{ases}$ (Fig. 13b), which gives us some confidence that large-scale differences with respect to the decoupling of P mineralization from the mineralization of C and N can be roughly captured by the model.

Direct measurements of bulk organic P mineralization using radio-isotopic approaches are only available for few soils. If up-scaled to yearly rates, measured organic P mineralization rates range
between $10^1$ to $10^3$ g P m⁻² yr⁻¹ (Bünemann, 2015; Wanek et al., 2019) which are orders of magnitude


higher than simulated by ORCHIDEE-CNP ($0.53 \pm 0.48$ g P m$^{-2}$ yr$^{-1}$; Fig. 13a). It has to be noted that lab measurements of P mineralization rates and potential phosphatase activity are conducted under optimal conditions, whereas in the field, temperature and water hamper enzyme activity for large parts of the year. This illustrates that the evaluation of model performance in representing soil organic P

mineralization remains a challenge since quantification of apparent rates under field conditions are currently missing.

**5.3 Global state of N and P budgets in natural and agricultural ecosystems**

**5.3.1 Openness and residence times of nutrients in natural ecosystems**

The openness of N and P cycles ($O_N$ and $O_P$) is a critical emerging property of an ecosystem which

measures the dependence of plant nutrient uptake on external nutrient sources (Cleveland et al., 2013; Wang et al., 2018). ORCHIDEE-CNP underestimates $O_N$ in tropical natural biomes, which is mainly due to lower, but more realistic, tropical BNF in ORCHIDEE-CNP compared to GOLUM-CNP used as a benchmark dataset (see 4.2.3). Modelled $O_P$ in natural biomes compares well with GOLUM-CNP except for central Africa (Fig. 8c, d), where we used a lower P deposition forcing than GOLUM-CNP.

Residence times of N and P ($\tau_N$ and $\tau_P$) are another critical property, not independent of openness though. ORCHIDEE-CNP simulates shorter $\tau_N$ and $\tau_P$ in tropical and temperate biomes compared to boreal ones, in line with GOLUM-CNP (Fig. 8). This indicates that ORCHIDEE-CNP is able to reproduce large-scale patterns in the nutrient residence time of biomes. Particularly, ORCHIDEE-CNP simulates higher $\tau_P$ for BOCF due to the higher biomass and soil P stocks (Fig. S12), related to the too

strong P limitation due to the too strong P immobilization in SOM in high latitudes (see section 5.1.3 and 5.4).

**5.3.2 Influences of agricultural activities on N and P fluxes**

The residence time of N and P in fertilized agricultural ecosystems is completely overridden by human input, so that model biases for those ecosystems should be related to how added nutrients are partitioned

between their incorporation in biomass and losses, and soil P accumulation for P through sorption. Total N fertilizer input and harvest N for cropland by ORCHIDEE-CNP are 125 Tg N yr$^{-1}$ and 75 Tg N yr$^{-1}$ in 2010, which gives a high N use efficiency for crop biomass production equal to 0.6 (NUE; defined as the ratio of harvest N yield to total N inputs). Modelled N harvest by ORCHIDEE-CNP is very close with the FAOSTAT-based estimates of 74 Tg N yr$^{-1}$ (Zhang et al., 2015), but our total N input

prescribed from Zhang et al. (2015) is lower than FAOSTAT-based estimate of 174 Tg N yr$^{-1}$. This difference is partly due to the different method to allocate fertilizer to crop and pasture areas by Zhang et al. (2015) (Conant et al., 2013; Lassaletta et al., 2014) and Lu and Tian (2017) (Monfreda et al., 2008). Further, BNF in croplands, about ~30 Tg N yr$^{-1}$ according to Bodirsky et al., (2012) and 39 Tg N yr$^{-1}$ for both cropland and pasture in (Bouwman et al., 2013b) was not accounted for in ORCHIDEE-

CNP but was accounted in Zhang et al. (2015). A lower estimation in total N input by ORCHIDEE-CNP leads to a higher NUE compared to Zhang et al. (2015) (0.42).

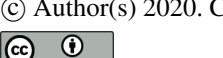


Averaged total P input for cropland over 2002-2010 is estimated to be 20.4 Tg P yr$^{-1}$, which is lower than the estimate by Lun et al., (2018) (25.5 Tg P yr$^{-1}$). This is because we used a lower manure P input (4.2 Tg P yr$^{-1}$) than Lun et al., (2018) (7.1 Tg P yr$^{-1}$) and omitted sludge P input (1.4 Tg P yr$^{-1}$ by Lun et al., 2018). Averaged modelled P harvest over 2002-2010 is 5.8 Tg P yr$^{-1}$, which is also lower than that of Lun et al. (2018); 11.7 Tg P yr$^{-1}$). The stoichiometry of crop harvest differs among crop types, while ORCHIDEE-CNP currently only distinguishes between three major crop types (maize, rice and wheat), which may explain the bias of P harvest whereas N harvest rates are in good agreement with independent estimates.

Modelled N and P leaching from croplands and pasture contributes 39% and 88% of global total N and P leaching in ORCHIDEE-CNP, respectively. The simulated global P leaching from managed land (3.3 Tg P yr$^{-1}$) is lower than the estimation by Lun et al. (2018) of 5.4 Tg P yr$^{-1}$) but higher than in Mekonnen and Hoekstra (2018) (0.6 Tg P yr$^{-1}$). This large range of published estimates is due to the different methodologies and the assumptions of the ratio of P loss to P input (Liu et al., 2018). For 895    example, Lun et al. (2018) assumed that a constant 12.5% of total P input was leached, while Mekonnen and Hoekstra (2018) estimated gridded P load to freshwater based on a much smaller erosion-runoff-leaching fraction (2.8% of total P input) for diffuse sources (Table S5).

**5.4 Relative N and P limitation based on leaf and soil stoichiometry**

Measurements show a decrease in foliar N:P ratios from low to high latitudes in natural ecosystems 900    (McGroddy et al., 2004; Reich and Oleksyn, 2004; Kerkhoff et al., 2005). This is seen as evidence for tropical vegetation being in general more P- than N-limited than extra-tropical vegetation (Reich and Oleksyn, 2004). The observed trend of foliar N:P ratios was not reproduced in the ORCHIDEE-CNP (Fig. 15) which simulated a flat foliar N:P latitudinal profile. In contrast to the majority of global models, where leaf N:P ratios are either prescribed (Goll et al., 2012) or vary within a PFT-specific 905    range (Wang et al., 2010), we assumed conservatively a globally uniform range to let the model freely calculate leaf N:P stoichiometry. It is not trivial to pin down the failure of the model to capture the latitudinal trend in leaf N:P ratios, which could be due to: 1) omitted variability in leaf P resorption efficiencies, which varies among biomes between 46%~66.6% (Reed et al., 2012), but was set to 65% in ORCHIDEE-CNP, 2) the simplistic parameterization of nutrient investment into different plant 910    tissues (see section 5.1.2), 3) and the omission of the diversity of nutrient acquisition pathways (e.g. mycorrhizal association) and rooting strategies (Warren et al., 2015). Testing new formulation for plant growth based on optimality principles (Kvakić et al., 2020) and the refinement of nutrient acquisition pathways (Sulman et al., 2017) are ways forward to improve the model.

Regarding soil stoichiometry, measurements show that tropical biomes have lower soil C:N and higher 915    soil C:P and soil N:P than temperate biomes (Tipping et al., 2016), echoing the pattern of leaf stoichiometry. ORCHIDEE-CNP fails in capturing these patterns (Figs 15 and 16). Modelled soil N:P and C:P for tropical forests are comparable to measurements but are too low in temperate forest, tropical and temperate grass, which is most likely related to a too strong nutrient immobilization in accumulating soil organic matter (Fig. S12) which tends to push systems into P limitation rather than N 920    limitation as $O_N$ is larger than $O_P$. In general, the spread in soil P concentration is well represented by





ORCHIDEE-CNP. The rudimentary representation of organic matter decomposition and the lack of nutrient effects on decomposers carbon use efficiency (see Zhang et al., 2018 for possible improvements, section 5.1.3) are likely contributing to the biases. New developments including explicit representation of decomposer communities and soil organic matter stabilization (Zhang et al., 2020) will be included in the next model version.

**5.5 Resource use efficiencies**

With the inclusion of nutrients, changes in the simulated use efficiencies of resources like water (WUE), light (LUE) and carbon (CUE) are expected. Indeed, the annual use efficiencies on biome-scale differ between ORCHIDEE-CNP and ORCHIDEE. In comparison to observation-based estimates, the inclusion of nutrient cycles tends to improve simulated LUE and CUE but not WUE.

Both ORCHIDEE-CNP and ORCHIDEE generally underestimates annual LUE for forest biomes which is due to a high bias in fAPAR in both models (28%-380% for ORCHIDEE, and 80%-173% for ORCHIDEE-CNP) (Fig. S13a, b). Although the bias in LUE for TRF is higher, the bias in GPP is largely reduced whereas the bias in fAPAR is similar in ORCHIDEE-CNP compared to ORCHIDEE (Fig. S13a, b), indicating general issues in ORCHIDEE with respect to how light is transferred within canopy in tropical forest. Both versions assume constant canopy light extinction coefficient of 0.5 (section 4.4), omitting variations among biomes due their distinctive canopy architectures (Ito et al., 2004). Improving this part of the model requires a canopy light transfer scheme that better accounts of canopy structure (Naudt et al., 2015) and the inclusion of different light components including diffuse incoming, scattered and direct light (Zhang et al. in prep).

ORCHIDEE-CNP simulated a lower WUE than ORCHIDEE, the latter one is closer to estimates based on eddy-covariance measurement networks (Fig. 17b). The improvement of WUE in TRF is related to improvements in GPP and ET, while the overestimation of WUE in coniferous dominated biomes by ORCHIDEE-CNP is related to an overestimation of GPP (Fig. S13c). The latter is likely a result of the application of a relationship between photosynthetic capacity and leaf nutrient concentration which is based on broadleaf species for all plant functional types (PFTs) in ORCHIDEE-CNP. Kattge et al. (2009) showed that coniferous PFTs have a ~40% lower carboxylation capacity for a given leaf nitrogen concentration than other PFTs. The omission of this could explain the bias in coniferous GPP in ORCHIDEE-CNP. Uncertainties in evaluation datasets hamper a more detailed evaluation of the variations of WUE among biome types.

We found that the inclusion of nutrient cycles improved the spatial variability in simulated CUE, but general biases remain (Fig. 17c), and uncertainties in observation-based estimates are large. Improvements are mainly found in temperate biomes (TEDF, TECF and TEG), indicating the allocation of GPP to respiration and biomass growth, which is controlled by nutrient availability, works reasonably well. ORCHIDEE-CNP underestimates CUE for tropical biomes (TRF and TRG) more strongly than ORCHIDEE, despite substantially reduced biases in NPP and GPP (Fig. S13d). However, we should be cautious in drawing conclusions considering the large uncertainty in MODIS CUE (He et al., 2018).





NUE, PUE on biome scale compare well to estimates (Fig. 18), indicating that ORCHIDEE-CNP is able
to simulate the coupling strength between C, N and P cycles. However, ORCHIDEE-CNP
underestimates PUE in tropical forests. A sensitivity analysis by GOLUM-CNP indicated that NUE and
PUE were most sensitive to the NPP-allocation fractions (especially to woody biomass) and foliar
stoichiometry (Wang et al., 2018). Therefore, we attribute the biases in PUE to the biases in foliar
stoichiometry (section 5.4) and to issues in plant internal P allocation in ORCHIDEE-CNP (section
5.1.2).

## 6 Concluding remarks

In this study, we evaluated the performance of ORCHIDEE-CNP in four aspects: 1) terrestrial C fluxes,
2) N and P fluxes and budget, 3) leaf and soil stoichiometry and 4) plant resource use efficiencies,
which is the most thorough evaluation of N and P dynamics in a LSM so far. In general, we found that
the model has some skills in reproducing the cycles of N and P in natural and managed systems. The
inclusion of nutrients improves the simulation of the sensitivity of plant productivity to increasing $CO_2$
and to inter-annual variation in precipitation. However, the nutrient-enabled version cannot capture the
current land carbon sink in the NH. This suggests that either the land carbon sink might be less a
consequence of the $CO_2$ fertilization effect, but of other processes that are currently not well resolved in
global models (e.g. biomass turnover, land management), or that ORCHIDEE-CNP underestimates the
ability of ecosystems to circumpass nutrient constraints on biomass built up under elevated $CO_2$.
We identified several shortcomings of the model, which are specific to nutrient cycling (lack of spatial
pattern in foliar stoichiometry, nutrient immobilization in soil organic matter, bias in N and P exports to
river catchments by leaching and the dynamics of soil inorganic P) as well as a general bias in LUE, C
sink in NH region, and inter-annual variability and seasonality of GPP. We propose the following focus
to improve ORCHIDEE in next model versions: 1) refine the canopy light absorb processes; 2) use
model-data assimilation frameworks (like ORCHIDAS) to better calibrate root phenology,
mineralization, nutrient resorption and growth allocation; 3) better represent soil processes related to
decomposition, stabilization of soil organic matter (e.g. Zhang et al., 2018, 2020) and inorganic P
transformation (e.g Helfenstein et al., 2020); 4) refine dynamics of biomass turnover and biomass
production efficiency including effects of forest management and climate. Continued improvements of
nutrient cycle representations will further reduce uncertainties in predicting land C sink under climate
change and rising atmospheric $CO_2$.

### Appendix

**A: Spin-up and pre-industrial simulations**

The historic simulation is preceded by three steps of spin-up to equilibrate plant and soil organic C, N
and P to pre-industrial conditions. The spin-up steps are forced by looped climate fields over the period
1901-1920, fixed pre-industrial atmospheric $CO_2$ (286 ppm) and land cover maps. We run the *spin-up 1*
over 200 years with forced C:N:P ratios of plant and soil, followed by two iterations of an analytic spin-





up (40 years for each iteration) (Vuichard et al., 2019). This facilitates the accumulation of soil organic C, N and P and approaches the equilibrium for plant productivity. We then run the *spin-up 2* from the last year of the *spin-up 1* with dynamic C:N:P ratios of plant and soils for 7k years, to reach a quasi-equilibrium for plant and soil N and P pools. In the last step, crop management (harvest and fertilizer) was activated for 200 years.

**B: Global Observation-based Land-ecosystems Utilization Model of Carbon, Nitrogen and Phosphorus (GOLUM-CNP)**

Global Observation-based Land-ecosystems Utilization Model of Carbon, Nitrogen and Phosphorus (GOLUM-CNP) v1.0 is a data-driven modeling of steady-state carbon, nitrogen and phosphorus cycles for present day (2001-2016) conditions. It combines the CARbon DAta MOdel fraMework
(CARDAMOM) data-constrained C-cycle analysis with spatially explicit data-driven estimates of N and P inputs and losses and with observed stoichiometric ratios. We extracted the following variables from GOLUM: global GPP and NPP, N and P in- and out- fluxes (e.g. BNF, N and P deposition, N and P leaching) on global and biome scale, biome-specific N and P resource use efficiencies, N and P openness, and N and P turnover rate. GOLUM-CNP only computed fluxes for non-cropland biomes. To
compare the global total stocks and flux budgets of C, N and P from GOLUM-CNP with those from ORCHIDEE-CNP, we multiply the stock and flux in each grid cell of GOLUM-CNP by the non-cropland fraction from the MODIS land cover map (version MCD12C1v006, https://lpdaac.usgs.gov/products/mcd12c1v006/) and sum over all the grid cells.

**C: Evaluation datasets for the C budget**

**C1 GPP and NPP from MODIS**

MODIS GPP is derived from climate and satellite data. It was estimated at 1 km spatial resolution from a light-use efficiency (LUE) model, as part of the operational MODIS algorithms (Running et al., 2004) using meteorological data from NASA Data Assimilation Office and detailed vegetation information (land cover and FPAR) derived from the Moderate Resolution Imaging Spectroradiometer (MODIS)
satellite from 2000 to present (Running et al., 2004; Zhao et al., 2005; Turner et al., 2006).

**C2 MTE GPP**

MTE GPP is a data-oriented product derived by extrapolating the GPP estimates from a network of flux-tower of 178 sites in space and time using climate data and remotely sensed fAPAR data from 1982 to 2008 (Jung et al., 2009). This latter monthly GPP product covers the period 1982-2011 at 0.5°
spatial resolution, but many of the underlying flux tower observations cover much shorter periods.

**C3 BESS-GPP**

Breathing Earth System Simulator (BESS) is a simplified process-based model that couples atmosphere and canopy radiative transfers, canopy photosynthesis, transpiration, and energy balance. This process-based model uses the MODIS Atmosphere products to calculate atmospheric radiative transfer. This
product provides gridded GPP on a monthly scale from 2000-2015 at 0.5° spatial resolution.





**C4 BETHY-NPP**

Biosphere Energy-Transfer Hydrology (BETHY)/DLR is a Soil-Vegetation-Atmosphere (SVAT) model to simulate photosynthesis of terrestrial ecosystems, which is operated at the German Aerospace Center (DLR). It was adapted from the BETHY scheme to be driven by remote sensing data (leaf area index (LAI) and land cover information) and meteorology (Wißkirchen et al., 2013). This product provides gridded global NPP on a monthly scale from 2000-2009 at 0.008° spatial resolution.

**C5 GIMMS-NPP**

GIMMS-NPP is a NPP product derived from the MODIS NPP algorithm driven by long-term Global Inventory Modeling and Mapping Studies (GIMMS) FPAR and LAI data (Smith et al., 2016). The uncertainty bounds were given by using a wide range of parameter combinations. This global satellite-derived NPP is available from 1982-2011 with a spatial resolution of 0.5° x 0.5°.

**C6 Net biome productivity from GCP and inversion data**

Estimates are based on 16 DGVMs involved in GCP Trendy v6 were forced by CRU-NCEP temperature, precipitation, and cloud cover fields (transformed into incoming surface radiation) based on observations and provided on a 0.5° × 0.5° grid and updated to 2016. We compared ORCHIDEE-CNP with Trendy v6 simulations instead of v7 because ORCHIDEE-CNP was part of the v7 ensemble.

We used three sets of $CO_2$ inversion data from CarbonTracker (van der Laan-Luijkx et al., 2017), Jena CarboScope (Rödenbeck, 2005), and CAMS (Chevallier et al., 2005). They all used atmospheric inversion methods (based on the same Bayesian inversion principles) and atmospheric $CO_2$ observations from various flask and in situ networks to constrain the location of the combined total surface $CO_2$ fluxes from all sources. The three systems used different transport models, which was demonstrated to be a driving factor behind differences in atmospheric-based flux estimates, and specifically for the global distribution (Stephens et al., 2007).

**D: Evaluation datasets for the N and P budget**

**D1 Biological Nitrogen fixation (BNF) from Peng et al. (2019)**

We used the global gridded BNF (both symbiotic and asymbiotic) from Peng et al., (2019) for 2001-2010. They calculated BNF as a function of soil N, LAI, labile inorganic soil P, temperature, etc (Wang et al., 2007; Houlton et al., 2008) which were constrained by the observation-based estimates of C:N ratios of the various plant pools, soil microbial biomass and organic matter under present conditions.

**D2 N and P loads to rivers on basin scale by GlobalNEWS2 model**

Global Nutrient Export from Water-Sheds (GLOBNEWS) model generates global spatially explicit of N and P exports by rivers based on a mass-balance approach for the land surface (watershed) and river system for year 2000 (Mayorga et al., 2010).



### D3 Inorganic P turnover rate

Soil solution P turnover (Km) were derived from isotopic exchange kinetic (IEK) experiments and aggregated on soil order level (Helfenstein et al., 2018).

### D4 Data-driven acid phosphatase activity

This global metric is an extrapolation of 126 European site observations of potential acid phosphatase activity from field samples by using back-propagation artificial network 12 environmental drivers as
predictors (Sun et al., 2020).

### E: Evaluation datasets for leaf stoichiometry and resource use efficiencies

### E1 Gridded leaf N:P ratios

Global maps of leaf N and P concentrations datasets (Butler et al., 2017) were derived from global plant trait database Bayesian modeling. We used 100 sets of estimates of leaf N and P concentrations
provided by Butler et al. (2017) to generate 10000 sets of global gridded leaf N:P ratio. Mean values and 25%~75% percentiles of leaf N:P ratios over those 10000 datasets were used to evaluate leaf N:P ratios by ORCHIDEE-CNP.

### E2 Remote-sensing fAPAP and PAR datasets

Global SeaWiFS Level-3 generates estimate of the Fraction of Absorbed Photosynthetically Active
Radiation (FAPAR) over the land surface through an optimized index which values are computed on the basis of top of atmosphere bidirectional reflectance factor values, as well as information on the geometry of illumination and observation (Gobron et al., 2006a, b). This dataset is available for period 1997-2006 with spatial resolution of $0.01^{o}$.

### E3 Evapotranspiration and GPP from MTE

MTE-ET is data-oriented product derived by extrapolating the flux-tower observations from 4678 sites in space and time using climate data and remotely sensed data from 1982 to 2008 (Jung et al., 2010). This latter monthly GPP product now covers the period 1982-2011 at 0.5° spatial resolution, but many of the flux tower observations only cover much shorter periods.

### F: Evaluation datasets for LAI

The GIMMS-LAI3g data were derived from the Global Inventory Modeling and Mapping Studies (GIMMS) Normalized Differential Vegetation Index (NDVI) using the neural network algorithm (Zhu et al., 2013). This monthly dataset is available for period 1982-2015 with spatial resolution of 1/12 degree.



**G: Evaluation datasets for forest above-ground biomass**

The GlobBiomass dataset of above-ground biomass (AGB) density (Figures S14b) were generated by combining spaceborne Synthetic Aperture Radar (SAR), LiDAR (Light Detection And Ranging) and optical remote sensing observations. This data is available for the year of 2010 and with a spatial resolution of 100m. The global forest above-ground biomass density from GEOCARBON project (Figure S14c) were generated from three existing datasets (Saatchi et al., 2011; Baccini et al., 2012;

Santoro et al., 2015). The AGB estimate at $0.01^{\circ}$ spatial resolution in the tropics was the weighted averages of data used in Saatchi et al. (2011) and Baccini et al. (2012), while that in the temperate and boreal forest corresponds to the biomass estimated in Santoro et al. (2015). These three datasets (i.e. Saatchi, Baccini and Santoro) were based on EO images acquired in different epochs (i.e. year 2000, 2007-2008 and year 2010). Simulated above-ground forest biomass carbon for the year of 2010 is

compared with observation-based datasets.

**H: Evaluation datasets for SOC**

**H1 Global Soil Dataset for use in Earth System Models (GSDE)**

GSDE provides global gridded dataset of soil organic C (SOC) with a spatial resolution of 30 seconds and soil depth of 2.3m. This dataset is constructed based on the Soil Map of the World and various

regional and national soil databases, including soil attribute data and soil maps (Shangguan et al., 2014).

**H2 SoilGrids**

SoilGrids provides global gridded dataset of SOC with a spatial resolution of 250m and soil depth of 2m. This dataset is extrapolated from over 230 000 soil profile from the WoSIS database by using state-of-the-art machine learning methods (Hengl et al., 2017).

**I: sensitivity of GPP anomalies to anomalies of mean annual temperature ($S_T$), annual precipitation ($S_P$) and shortwave radiation ($S_R$)**

In this study, we estimated the response of GPP to climate variability by employing the multiple regression approach from Piao et al. (2013):

$$y = S_T x_T + S_P x_P + S_R x_R + \varepsilon$$

where y is the de-trended anomaly of GPP, $x_T$ is the detrended mean annual temperature anomaly, $x_P$ is the de-trended annual precipitation anomaly, and $x_R$ is the detrended mean annual radiation anomaly. The fitted regression coefficients $S_T$, $S_P$, $S_R$ define the apparent GPP sensitivity to inter-annual variations in temperature, precipitation and radiation, respectively, and $\varepsilon$ represents the residual error term.





**Code and data availability**

The source code is freely available online via the following address: http://forge.ipsl.jussieu.fr/orchidee/wiki/GroupActivities/CodeAvalaibilityPublication/ORCHIDEE-CN-P_v1.2_r5986 (Goll, 2020). Please contact the corresponding author if you plan an application of the model and envisage longer-term scientific collaboration.

Primary data and scripts used in the analysis and other supplementary information that may be useful in reproducing the author's work can be obtained by http://dx.doi.org/10.17632/f54v9zcgbf.1

**Supplement.** The supplement related to this article is available online at: (a doi will be provided before final publication).

**Author contributions.** YS and DSG carried out the simulation of ORCHIDEE-CNP. YS, JC, JH, VN, YW
and HY analyzed the model outputs. YS, DSG, and PC prepared the paper with contributions from all coauthors.

**Competing interests.** The authors declare that they have no conflict of interest.

**Acknowledgements.** YS, DSG, PC, HZ and YH are funded by the IMBALANCE-P project of the European Research Council (ERC-2013-SyG610028). JH is supported by the Swiss National Science
Foundation (Project number 200021_162422).

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



**Table 1 Main information on datasets used for global evaluation of ORCHIDEE-CNP.**

| Dataset | Variable | Resolution | Period | Uncertainties | References |
|---|---|---|---|---|---|
| MODIS | GPP, NPP, CUE | 1km | 2000-2015 | Bias against local measurements for GPP and NPP | Running et al., 2004; Zhao et al., 2005; Turner et al., 2006; |
| MTE | GPP, WUE | 0.5° | 1982-2011 | 25 ensemble trees for GPP and ET respectively | Jung et al., 2009; Jung et al., 2011 |
| BESS | GPP | 0.5° | 2001-2015 | Bias against local measurements | Ryu et al., 2011; Jiang and Ryu, 2016 |
| BETHY | NPP | 0.008° | 2000-2009 | - | Tum et al., 2016; Wißkirchen et al., 2013 |
| GIMMS | NPP | 0.5° | 1982-2015 | Using different climate inputs | Smith et al., 2016 |
| Trendy v6 | NBP | 0.5° | 1959-2016 | 1-sigma standard deviation | Sitch et al., 2013 |
| JENA_inversion | NBP | 1° | 1985-2016 | - | Rödenbeck et al., 2003 |
| CMAS inversion | NBP | 1.875°x3.75° | 1979-2016 | - | Chevallier et al., 2005 |
| Ctracker inversion | NBP | 1° | 2001-2016 | - | van der Laan-Luijkx et al., 2017 |
| Peng-BNF | BNF | biome | 2001-2009 | - | Peng et al., 2019 |
| Sullivan-BNF | BNF | biome | 1999, 2009 | - | Sullivan et al., 2014 |
| Mayorga | N & P leaching | polygon | 2000 | - | Mayorga et al. 2010 |
| Helfenstein | Km | Soil order | - | - | Helfenstein et al., 2018 |
| Sun | Pasae activity | 10km | - | - | Sun et al., 2020 |
| GOLUM-CNP | C, N and P fluxes, N and P openness and turnover rate, PUE, NUE | 0.25° | 2001-2010 | - | Wang et al., 2018 |
| Global SeaWiFS Level-3 data and MTE-GPP | LUE | 0.01° | 1997-2006 | - | Gobron et al., 2006a, b |
| Butler | Leaf N: P ratio | 1km | | 100 estimates by Bayes method | Butler et al., 2017 |
| Site leaf measurements | Leaf N:P ratio | site | - | - | Kerkhoff et al., 2005; McGroddy et al., 2004; Reich and Oleksyn, 2004 |
| Tipping | SOM C, N and P | site | - | - | Tipping et al., 2016 |
| Site measurements of NUE and PUE | NUE and PUE | site | - | - | Gill and Finzi, 2016 |



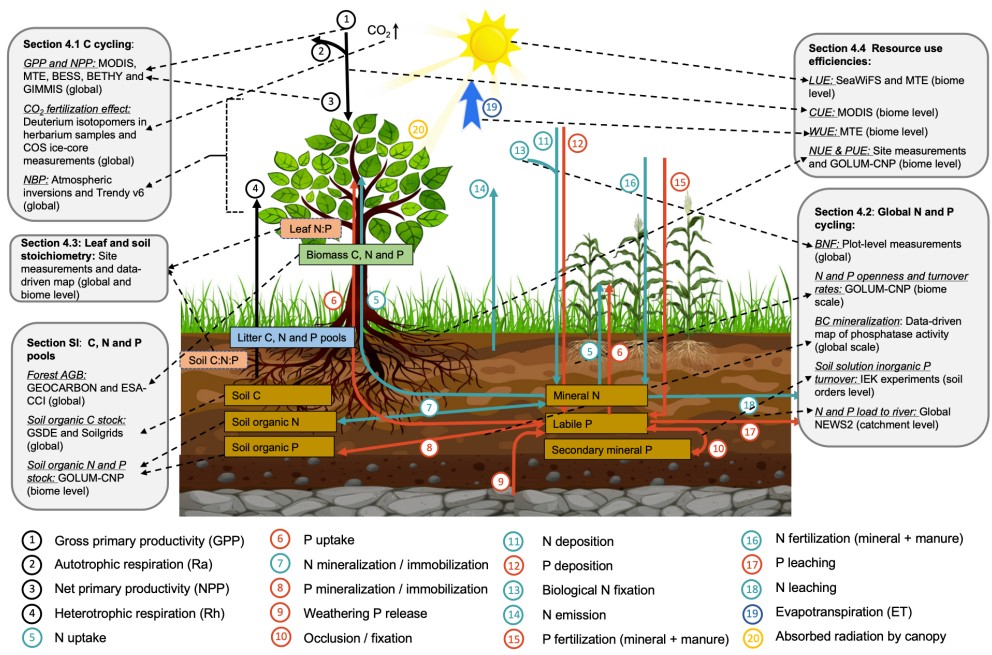


**Figure 1: Schematic of C, N and P cycles considered in ORCHIDEE-CNP. The evaluated variables and corresponding evaluation datasets are also shown in grey boxes.**



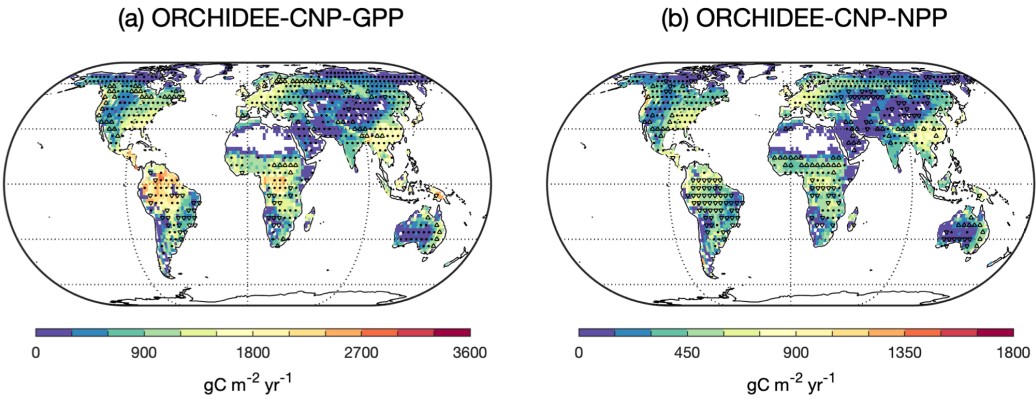

**Figure 2: Comparisons of global pattern for plant primary productivity between ORCHIDEE-CNP and data-driven products (MODIS-GPP, MTE-GPP, BESS-GPP, MODIS-NPP, BETHY-NPP, GIMMS-NPP). Black point, '∆' and '∇' indicates the GPP or NPP simulated by ORCHIDEE-CNP lie within the ranges / higher than the upper limits / lower than the lower limits of estimations by data-driven products, respectively. The ranges of data-driven GPP and NPP involves uncertainties within each product and among all products.**







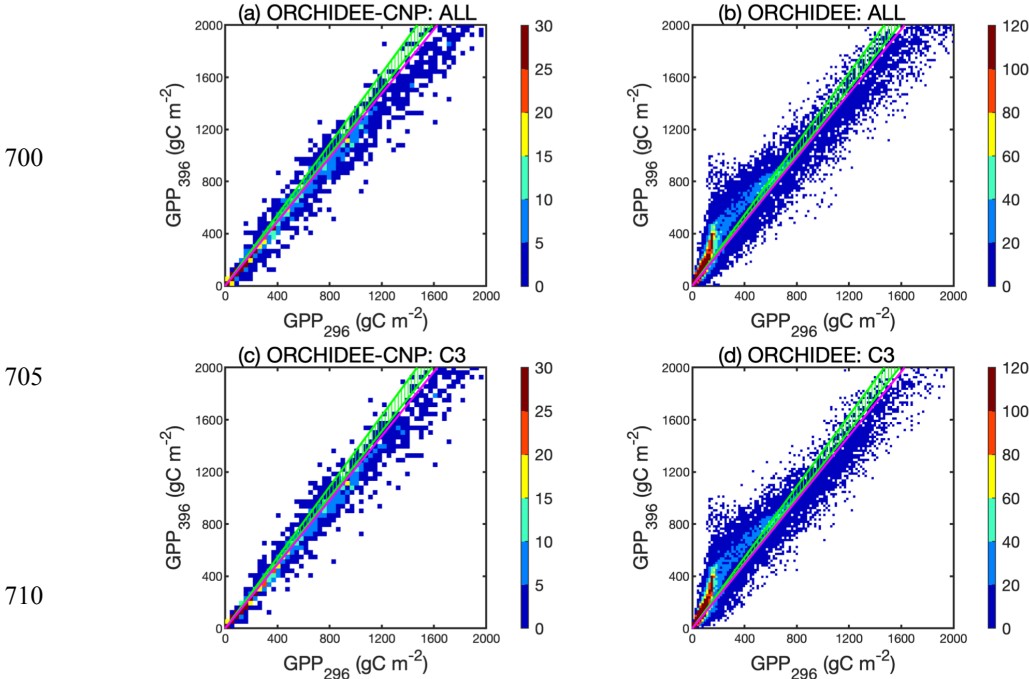

**Figure 3: Comparisons between pre-industrial GPP with atmospheric CO$_2$ concentration of 296 ppm (*GPP$_{296}$*) and current GPP with atmospheric CO$_2$ concentration of 396 ppm (*GPP$_{396}$*) for all natural plants (a, b) and natural C3 plants (c, d) by ORCHIDEE-CNP (a, c) and ORCHIDEE (b, d). The color scale shows the point density. Different point density and patch size for ORCHIDEE and ORCHIDEE-CNP are due to the different spatial resolution (2$^o$ x 2$^o$ for ORCHIDEE-CNP and 0.5$^o$ x 0.5$^o$ for ORCHIDEE). The ratio between *GPP$_{396}$* and *GPP$_{296}$* indicates the CO$_2$ fertilization effects (E$_{CO2}$). Green dashed areas indicate the observed E$_{CO2}$ from Campbell et al (2017)'s COS records. Rose lines indicate the observed E$_{CO2}$ from Ehlers et al (2015)'s.**



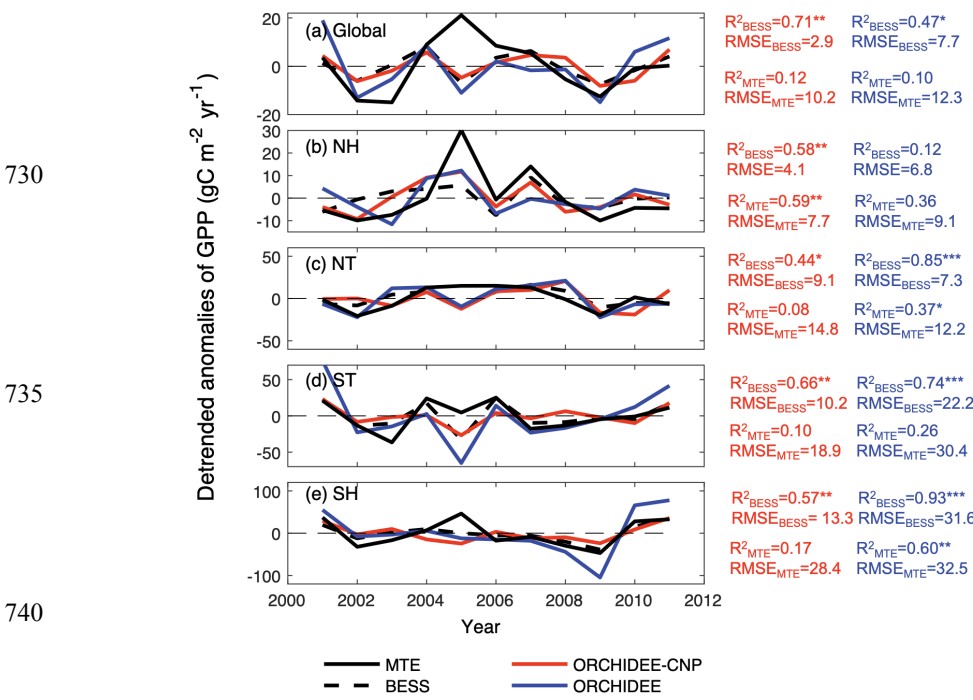

**Figure 4: De-trended anomalies of GPP by BESS-GPP, MTE-GPP, ORCHIDEE and ORCHIDEE-CNP for globe (a), the northern hemisphere (30°N-90 °N; b), north tropical (0°-30°N; c), south tropical (0°-30 °S; d) and the southern hemisphere (30°S-90 °S; e). Inter-annual correlations ($R^2$) and root mean square deviation (RMSE) between modelled GPP and BESS-GPP ($R^2_{BESS}$ and $RMSE_{BESS}$) and MTE-GPP ($R^2_{MTE}$ and $RMSE_{MTE}$) for ORCHIDEE-CNP (red) and ORCHIDEE (blue) are also shown. \*\*\* indicates p<0.001, \*\* indicates 0.001<p<0.01, \* indicates 0.01<p<0.05.**










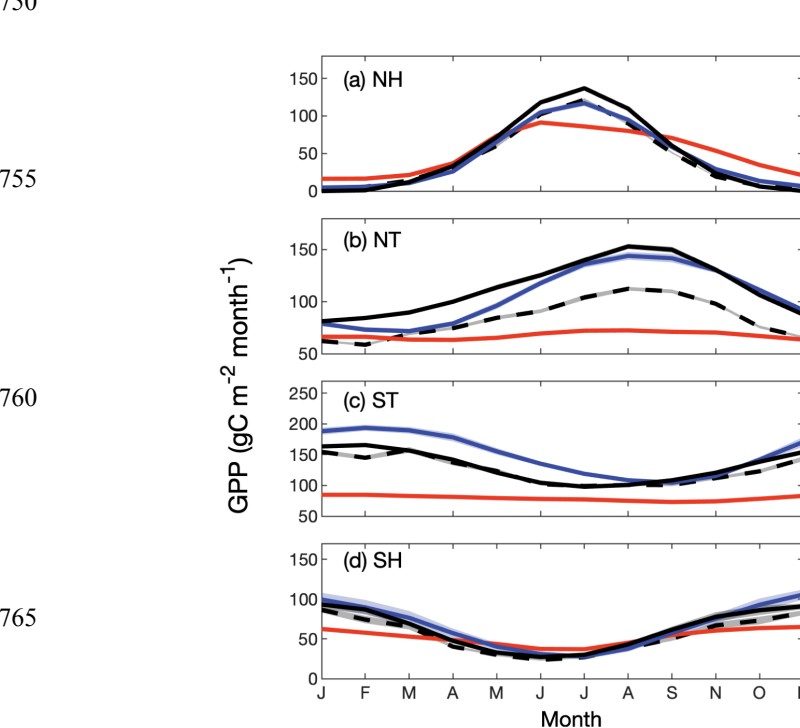

**Figure 5: Seasonal cycles of GPP by BESS-GPP, MTE-GPP, ORCHIDEE and ORCHIDEE-CNP for the northern hemisphere (30°N-90 °N; a), north tropical (0°-30°N; b), south tropical (0°-30 °S; c) and the southern hemisphere (30°S-90 °S; d).**


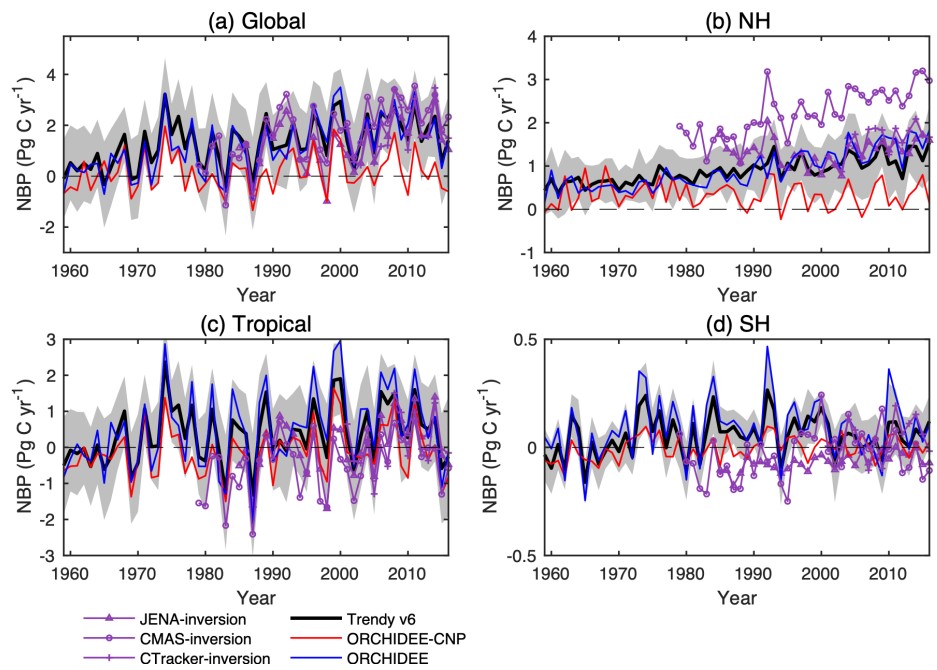


**Figure 6: Comparisons of net biome productivity (NBP) between ORCHIDEE-CNP, ORCHIDEE, GCP and inversion dataset for (a) globe (b) the northern hemisphere (30°N-90 °N), (c) tropical (30°S-30 °N), and (d) the southern hemisphere (30°S-90°S). Grey shading indicates ±1σ uncertainty of DGVM results for Trendy v6.**


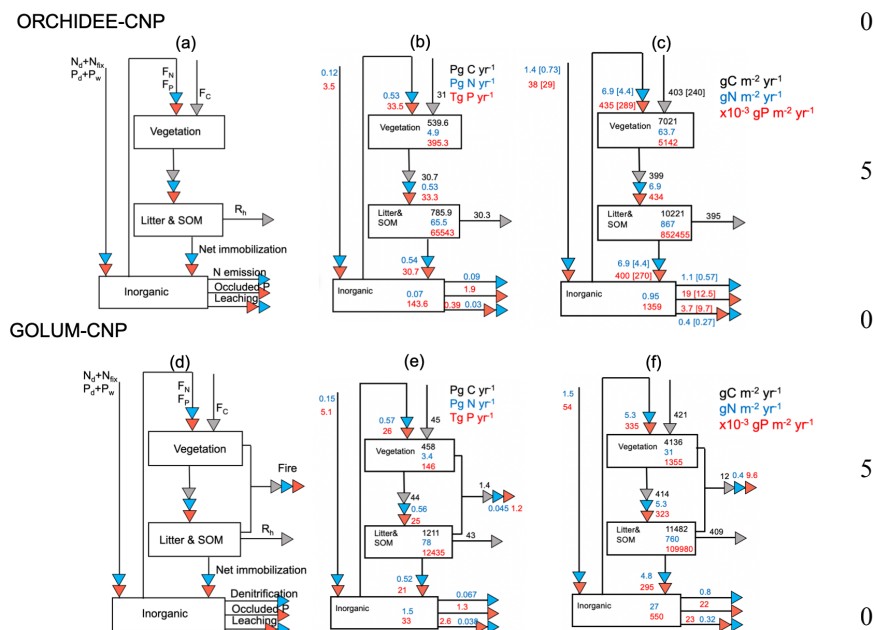

**Figure 7: Flow chart of total global fluxes (c, f) and mean flows per area (b, e) of C, N and P in natural biomes for GOLUM-CNP (e, f) and ORCHIDEE-CNP (b, c). Panel (a) and (d) show the schematic representation of C, N and P fluxes for ORCHIDEE-CNP and GOLUM-CNP. GOLUM-CNP stands for Global Observation-based Land-ecosystems Utilization Model of Carbon, Nitrogen and Phosphorus (GOLUM-CNP) v1.0, which is a data-driven model of steady-state C, N and P cycles for present day (2001-2016) conditions. Numbers in square brackets indicate the standard deviations for accounting the spatial spread of C, N and P fluxes.**





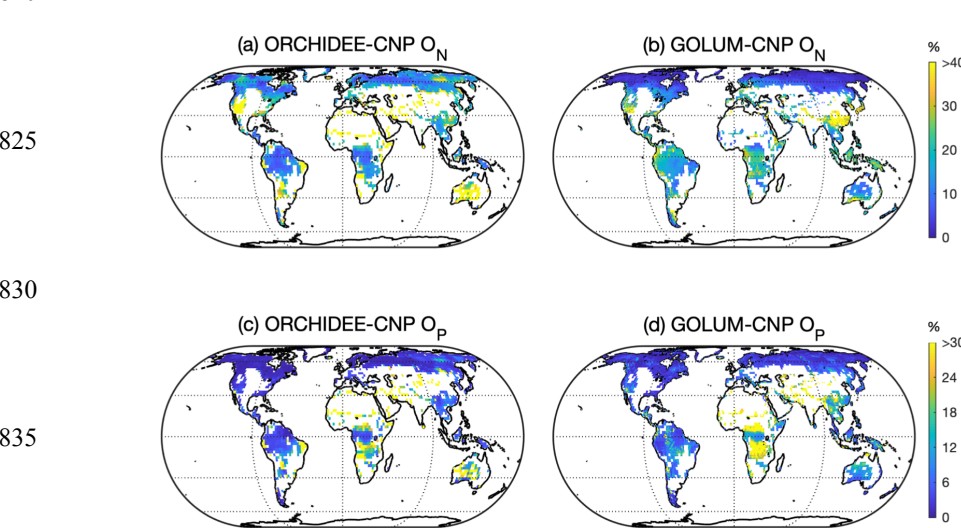




**Figure 8: Global pattern of N ($O_N$, a-b) and P openness and ($O_P$, c-d) simulated by ORCHIDEE-CNP (a, c) and GOLUM-CNP (b, d). Pixels with managed lands >50% in ORCHIDEE-CNP were masked. Same area was masked from the pattern of $O_N$ and $O_P$ for GOLUM-CNP.**









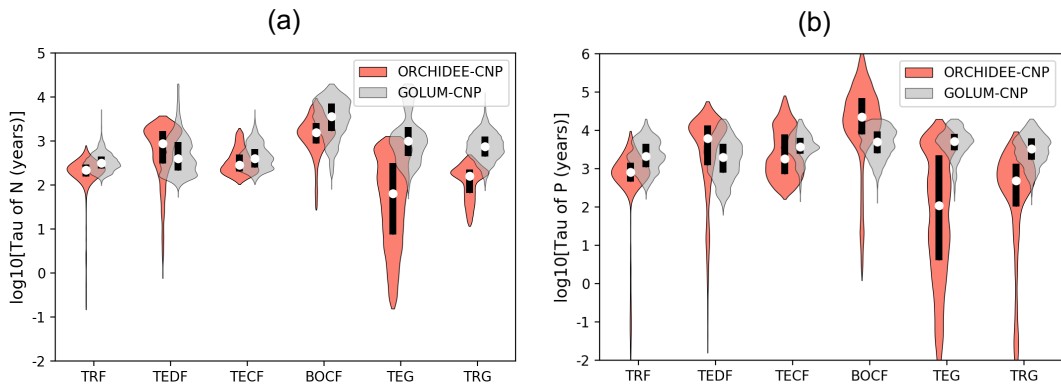

**Figure 9: Residence time (c, d) of N and P cycles for 6 biomes: tropical rainforest (TRF), temperate deciduous forest (TEDF), temperate coniferous forest (TECF), boreal coniferous forest (BOCF), temperate grass (TEG) and tropical grass (TRG).**




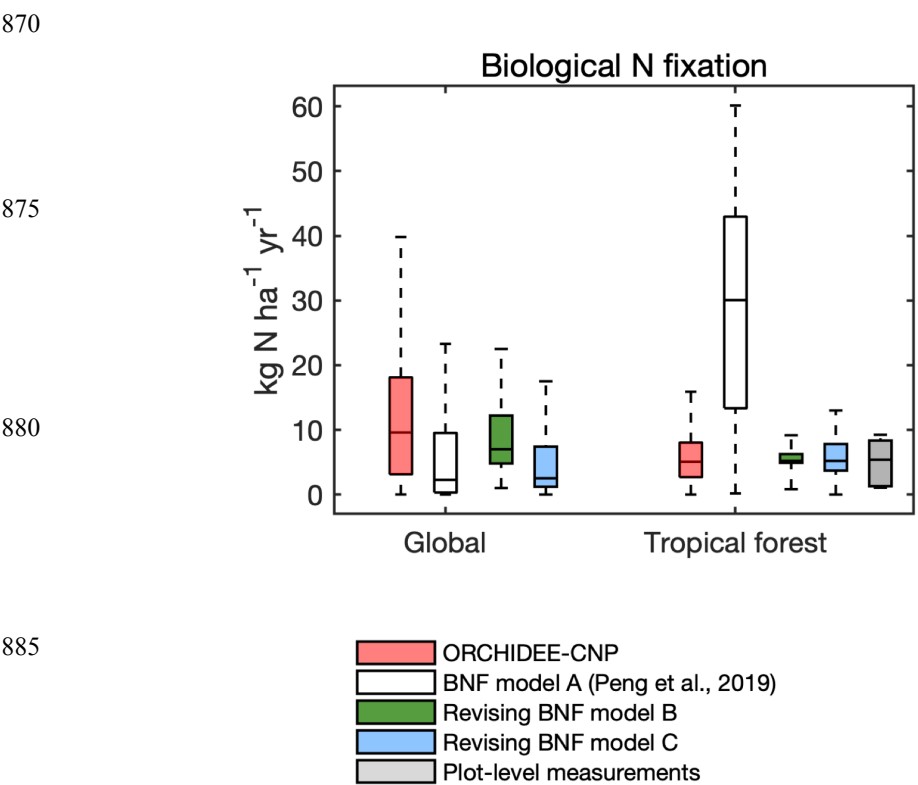


**Figure 10: Global BNF and tropical forest BNF simulated by ORCHIDEE-CNP, BNF model A by Peng et al. (2019), and revising BNF model B (Cleveland et al., 1999) and C (Wang and Houlton, 2009) based on plot-level measurements in tropical forests by Sullivan et al. (2014) (grey box).**




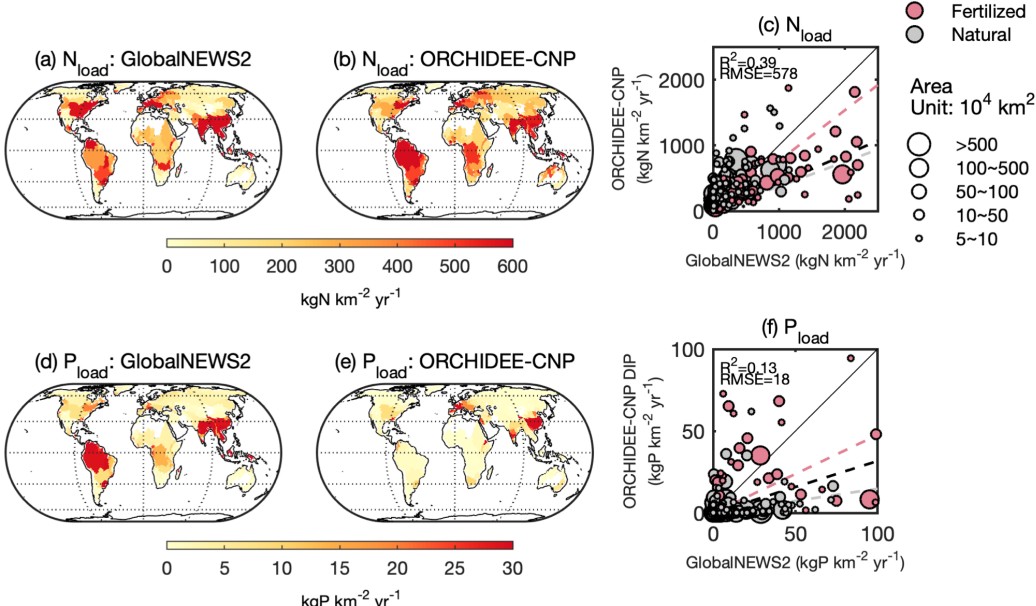

**Figure 11: Global pattern of N ($N_{load}$, a-c) and P loads ($P_{load}$, d-f) from land to rivers for basins with area larger than 50,000 km$^2$. (c) and (f) show the comparisons of N and P leaching between ORCHIDEE-CNP and GlobalNEWS2 (Mayorga et al., 2010). Black line indicates 1:1 line. Pink dashed line indicates the linear regression line for basins with fertilized basin (N fertilization higher 1 gNm$^{-2}$yr$^{-1}$ or P fertilization higher than 0.5 g Pm$^{-2}$yr$^{-1}$), while grey dashed line indicates the linear regression line for basins with natural basin (N fertilization lower 1 gNm$^{-2}$yr$^{-1}$ or P fertilization lower than 0.5 g Pm$^{-2}$yr$^{-1}$). Black dashed lines indicate the linear regression line for all basins. R$^2$ and RMSE refer to the coefficient of determination and root-mean-square error between estimations of ORCHIDEE-CNP and estimates from GlobalNEWS2 (Mayorga et al., 2010) for all basins.**








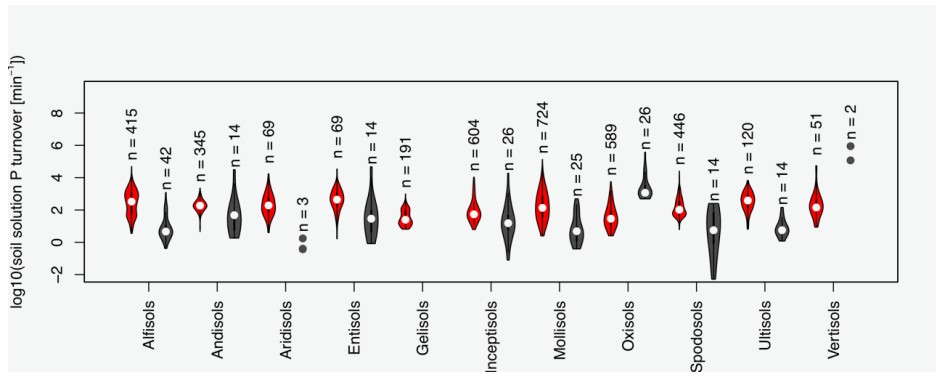

**Figure 12: Comparisons of soil solution inorganic P turnover rates by soil order between ORCHIDEE-CNP and measurements (Helfenstein et al. 2018).**

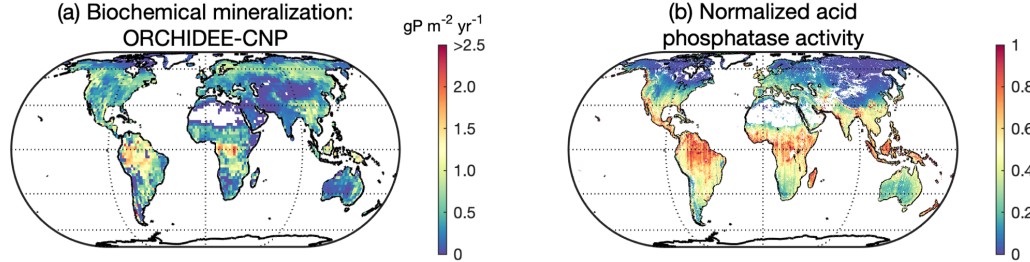

**Figure 13: Global pattern of biochemical mineralization by ORCHIDEE-CNP (a) and normalized acid phosphatase activity (b).**

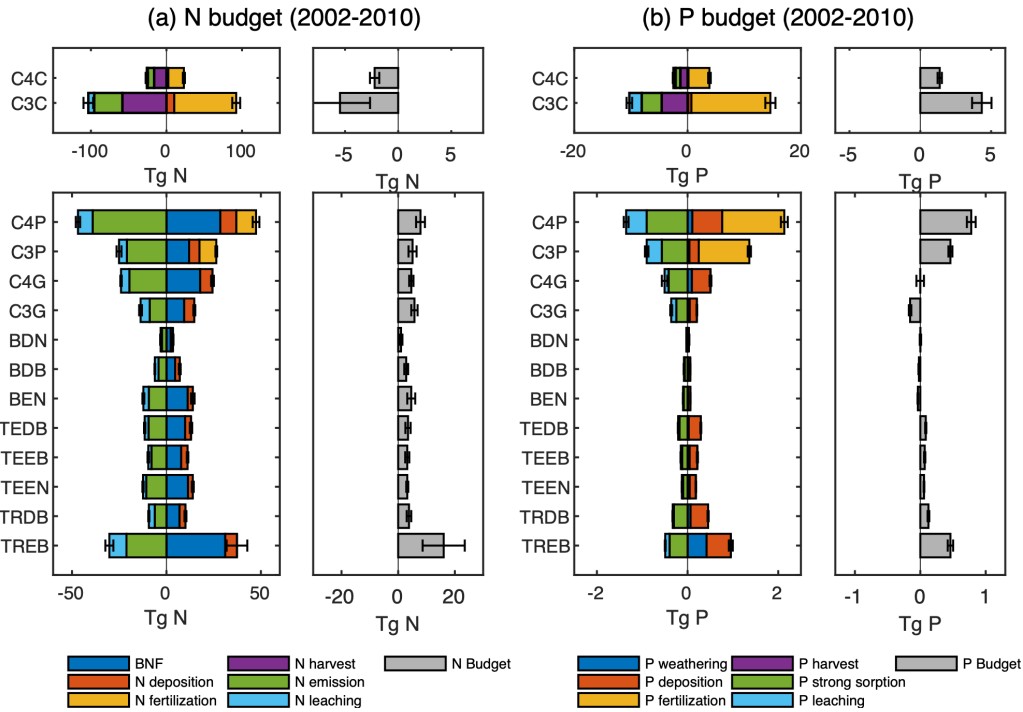

**Figure 14: Total N and P budgets for each biome. Error bars indicate the standard deviation over 2002-2010. TREB: tropical evergreen broadleaf forest; TRDB: tropical deciduous broadleaf forest; TEEN: temperate evergreen conifer forest; TEEB: temperate evergreen broadleaf forest; TEDB: temperate deciduous broadleaf forest; BEN: boreal evergreen conifer forest; BDN: boreal deciduous conifer forest; C3G: C3 grassland; C4G: C4 grassland; C3P: C3 pasture; C4P: C4 pasture; C3C: C3 cropland; C4C: C4 cropland.**




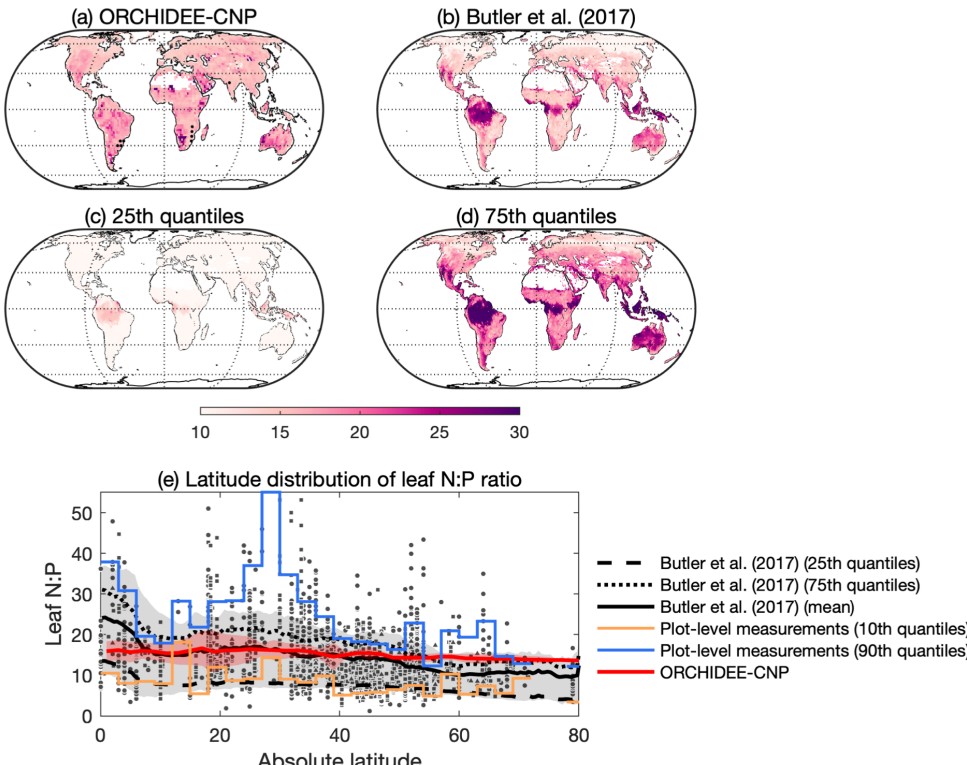

**Figure 15: Comparisons of leaf N:P ratio between ORCHIDEE-CNP, data-driven estimates and observations. (a) is the global pattern of mean leaf N:P ratio over 2001-2016 for ORCHIDEE, (b) is for mean leaf N:P in Butler et al. (2017). (c) and (d) are 25% and 75% percentile of leaf N:P ratio by Butler et al. (2017), respectively. Dots in (a) indicate the area with leaf N:P ratio of**

**ORCHIDEE-CNP falling into 25%~75% percentiles of Butler et al., (2017)'s estimation. (d) is the latitude distributions of leaf N:P ratio for ORCHIDEE-CNP, Butler et al (2017)'s estimation and site measurements. Red shared area indicates the uncertainty from latitudinal spreads of leaf N:P ratio for ORCHIDEE-CNP. Grey shaded area indicates the uncertainty from both the estimations and latitudinal spreads for Butler et al., (2017). Blue and yellow lines indicate the 10% and 90% percentiles of measured leaf N:P ratios in each bins of 3o latitude, respectively.**









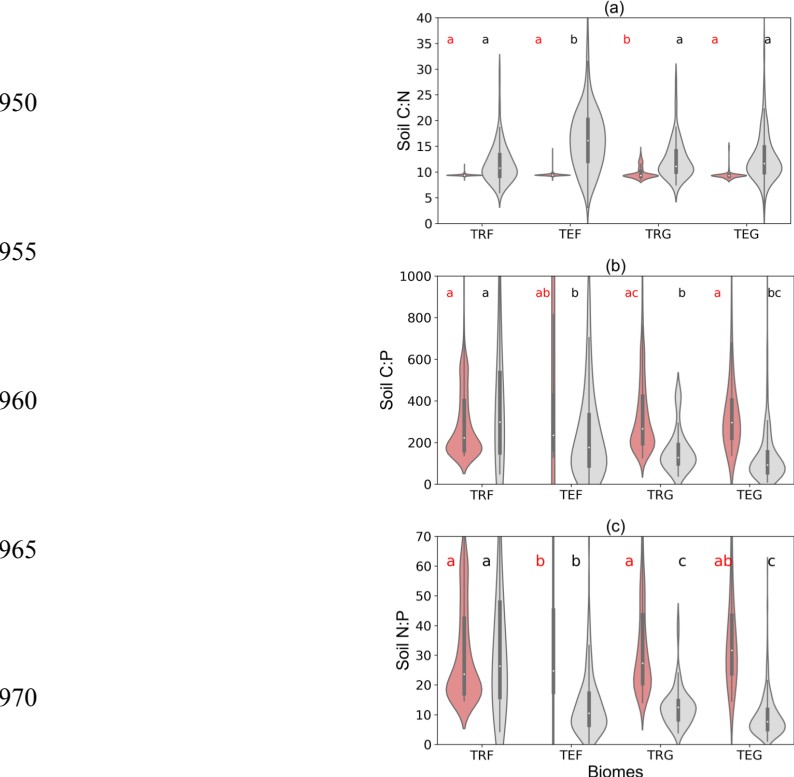

**Figure 16: C:N, C:P and N:P ratios of soil organic matter by ORCHIDEE-CNP and plot-level measurements by Tipping et al. (2016) for 4 biomes: tropical forest (TRF), temperate forest (TEF), tropical grass (TRG) and temperate grass (TEG). Soil C:N:P ratios for ORCHIDEE-CNP are calculated for total soil pool includes soil passive, slow and active pools, while measurements by Tipping et al. (2016) are for soils of 0-60 cm depth.**












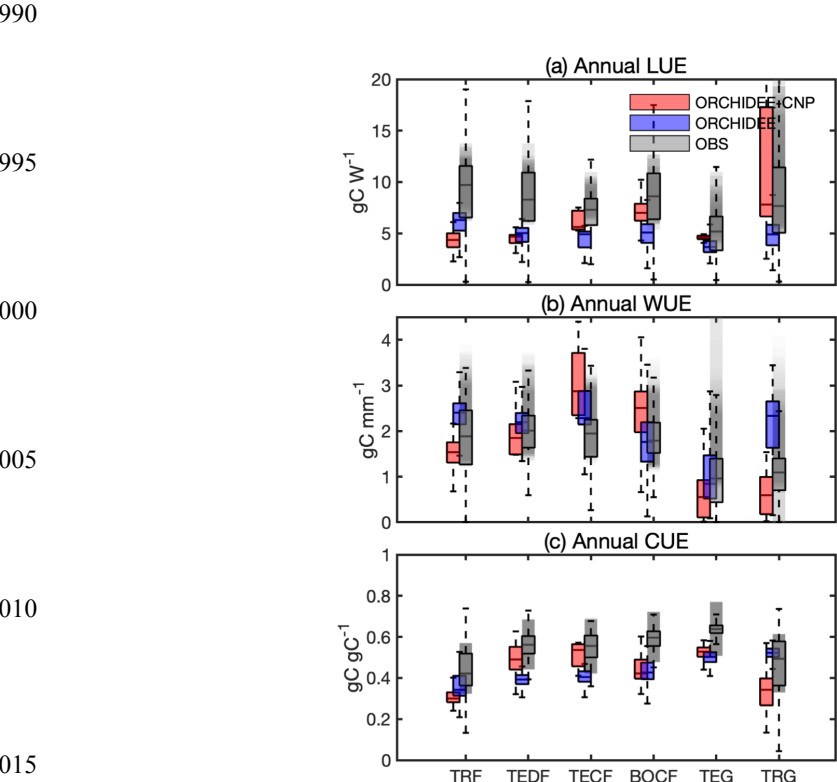

**Figure 17: Comparison of annual resource use efficiencies between ORCHIDEE-CNP, ORCHIDEE and satellite-based estimations for 6 biomes: tropical rainforest (TRF), temperate deciduous forest (TEDF), temperate conifer forest (TECF), boreal conifer forest (BOCF), temperate grass (TEG) and tropical grass (TRG). Black boxes indicate the satellite-based estimations (referenced), and the grey shaded areas indicate the uncertainties of resource use efficiencies given by referenced estimations, which involves uncertainties for multi-estimations and spatial variability for each estimation.**







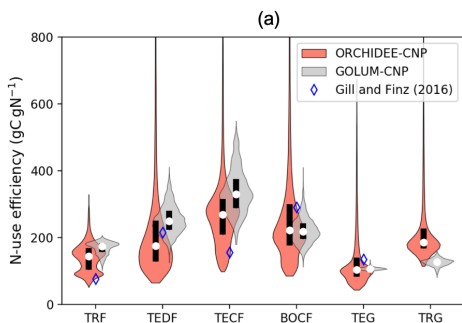
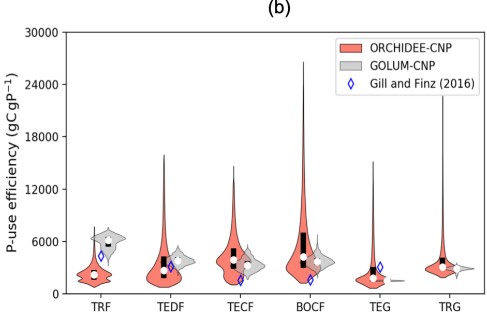


**Figure 18: Nitrogen use efficiency (NUE; a) and phosphorus use efficiency (PUE; b) by ORCHIDEE-CNP, GOLUM-CNP and observations (Gill and Finzi, 2018) for 6 biomes: tropical rainforest (TRF), temperate deciduous forest (TEDF), temperate conifer forest (TECF), boreal conifer forest (BOCF), temperate grass (TEG) and tropical grass (TRG).**