# Peer review of "Global evaluation of nutrient enabled version land surface model ORCHIDEE-CNP v1.2 (r5986)"

_Geoscientific Model Development, 2020_

## Referee Comment (RC1) · Anonymous Referee #1 · 5 Oct 2020

Sun et al., performed a comprehensive evaluation of the nutrient-enabled version of the ORCHIDEE model (ORCHIDEE-CNP). The evaluations were made for biosphere carbon fluxes, N and P cycles, leaf and soil conditions and plant resource use efficiencies. Based on this, the authors were able to provide clear recommendations for future development.

The extensive set of observational data that the authors use, together with the evaluation of different metrics is very powerful. The work provides a complete picture of the model performance. My main concern, however, is that due to the comprehensiveness of the work it is at the same time difficult to grasp the main messages when reading the results and discussion section. When I read the manuscript for the first time I was overwhelmed by all the data and comparisons. I did not get a clear picture of the key

messages in between the discussion of all the different metrics. The length of the manuscript contributes to this as well (27 pages of text, 18 figures and a table, plus several supplementary documents). Instead of showing comparisons of all available observational datasets, the authors could choose to only show those figures that help to illustrate their conclusion, and move other comparisons to the supplementary material and discuss them only shortly. For example, the authors discuss in sect 5.1.2 that it is difficult to falsify one model version over another based on the comparison of the de-trended anomalies in Fig 4, due to uncertainties in the observed GPP. As this figure did not contribute much to their final conclusion, the authors could choose to leave Fig. 4 out of the main manuscript. I would advise the authors to have a critical look at their figures, and the messages that they convey, to see which figures are really key to bring their message forward in order to condense (and thereby improve) the manuscript.

Moreover, section 2 "Model description" gives an overview of the modifications that were made compared to ORCHIDEE-CNP v1.1, but it does not provide an overview of the nutrient flows of N and P. A brief description of steps in the N and P cycles would be helpful to understand the processes in this nutrient-enabled model version of ORCHIDEE. When such processes are introduced this would also help to understand the evaluation with observations later on. A discussion of Fig. 1 could serve this, as it is a nice illustration of such processes, but this figure is currently hardly discussed in the text.

I provide specific comments below:

P2, Line 49: "this direction of future carbon storage", what direction? P2, line 58: e? P2, line 70: "should look for" = needs. P2-3, line 75-81: Shorten or break up this sentence for readability. P4, line 118: give full name of SOM. P4, line 140-142: mention here the resolution of the ORCHIDEE run (0.5 degree) that differs from the 2.0 degree for ORCHIDEE-CNP. Otherwise this is only mentioned in the caption of Fig. 3, but it is relevant information. P8, line 277-279: check correctness of this sentence. P8, line 287: its uncertainties were calculated P8, line 290: based on P8, line 291: remove

"speaking" P9, line 311: twice higher = twice as high P9, line 316-318: The markers in Fig. 2b for temperate and western Europe don't seem to show this? P10, line345-347: Over the whole range, Eco2 seems to be quite similar for ORCHIDEE-CNP and ORCHIDEE. Is the ORCHIDEE Eco2 higher only because it has more data points in the lower GPP range where Eco2 is clearly higher than those based on Campbell et al. (2017) and Ehlers et al. (2015)? What is the role of the different resolution of ORCHIDEE-CNP and ORCHIDEE here? Does the resolution explain that there are data points in the ORCHIDEE plot below ~400 GPP-296 and in between ~300-1000 GPP-396, that are not there for ORCHIDEE-CNP? P11, line 379: refer to Fig. 6a here already P11, line 381: "... but is close to JENA-inversion estimate during this period". Can you give the value for the CTracker atmospheric inversion? P11, line 381: Global simulated NBP from ORCHIDEE-CNP.... P11, line 392: avoid the word "cause" here. Better say something like: "Therefore, the underestimation of the global C sink in ORCHIDEE-CNP during the last decades is primarily due to a lower C sink in the NH." P12, line 13: abbreviation BNF was introduced already earlier. P13, line 457-468: this discussion of literature values is longer than needed, please shorten. P15, line 537:-538 it is not entirely clear to me how I can see the net N accumulation of 51.5 Tg N yr-1 from Fig 14a?

Discussion: as there were so many results presented, it is important to reference to a figure or paragraph from the results section that evidences your statements. E.g. in line 631-634; 635-638.

P18, line 643: "all models", you mean biosphere models? P19, line 676-677: but aren't ORCHIDEE-CNP and ORCHIDEE forced with the same meteorology, and thus the same temperature, precipitation and radiation? P20, line 691: you mean Fig. S7? P20, line 711: "...of the land C sink...", you mean "the size of the land C sink"? P21,line 768: is "capital" the right word here? Maybe use "pools"? P23, line 825: "due to that..." = "that is because .... " P23, line 831: model = models P23, line 830-834: shorten this sentence for readability.

Appendices and Supplementary Material: It is confusing that the manuscript contains Appendices ánd two supplementary documents. It would be good if all such supplementary material is combined into a single document.

Tables and Figures: Table 1: CMAS inversion = CAMS inversion? Fig 3, legend: Rose lines = Pink lines? Fig 7, complex, consider removing it to the supplementary material. Fig 10, Besides mentioning "model B" and "model C", also give the reference in the legend (like for model A, Peng et al., 2019 is mentioned). Fig 12, 16, what is red and what is black? Fig 12, is n from the model the number of gridcells with that soil type? Fig 9, 12, 16, 18 add in the legend what the width of the bars indicates. Fig 16, what do a, b, c in the figures mean?

---

## Referee Comment (RC2) · Anonymous Referee #2 · 11 Oct 2020

The work by Sun et al. is impressive in the sense that many analysis are taken to understand the Orchidee-CNP model. But on the other hand the paper lacks a clear flow of arguments.

1. The reason why every time the models need to be more detailed is that we are not satisfied by the performance of the old models. If we only focus on carbon and water (WUE) then we clearly see problems in the dynamics, sinks and sources, which was the reason to include Nitrogen and now also the Phosphorus cycle. However, if one the main conclusions is that the current version of this model is unable to simulate carbon sinks, then the choices of expanding the model need to N and P should be much more discussed in detail. As long as we can not model the carbon cycle, what kind of implications has this on the N and P cycle? If there is a large problem in land

carbon sink estimates, then I would like to see the consequences to all other coupled processes, including water. If this is large, then this should be solved first or we should simply conclude that there is too less understanding to couple the models as proposed in a global model. An evaluation as proposed by the authors of this version of the model doesn't help us in answering this problem.

2. A second major concern is therefore that the evaluation is far too broad while missing the in depth analyses. The number of figures are too many and jumping from one type of comparisons to another: a. On one hand you are showing the dynamics, but then I would like to have much more information on understanding the drivers. For instance how much dependent is the dynamics of P and N on the P and N deposition? There are studies who have shown in other models that this N Deposition is one of the main drivers. b. Then you make some snap shots of global patterns, while later on you focus on different ecoregions and then different soil types and then on vegetation with different photosynthetic pathways. It would be very helpful to structure this far better and to integrate those aspects.

3. A third major concern is the for me random way of comparing the results. I found the comparison with only ORHIDEE-C not very convincing. Why not comparing to the average performance of the land models as done in TRENDY? There are other global model results as well on C and N. For instance LPJ guess.

Then I have a couple of smaller remarks, but are not extensively as in next version I would expect that parts of analysis will not be reported anymore and need to be seen as a number of examples: 4. L402: why do you have a smaller natural land cover? Is it a problem from ORCHIDEE or from GOLUM-CNP. Is it then still useful to compare? 5. Comparing global news N-leaching: other forcings → this doesn't help us in understanding the role of the different mechanisms → can you also compare it with similar forcings? If not, is it still valid to include this comparison? 6. How did you downscale from HYDE3.2 to 1x1 km? Did you use the same allocation rules as done by Klein Goldewijk for the 30 minute resolution? 7. L890: N and P leaching: the current

problem of understanding N and P leaching is large: there are all kind of confounding factors that determine these leaching rates which are in the end extremely important to understand water quality and functioning of the system. In the current paper I can not find this sensitivity back. Leaching is highly dynamic due to fire, soil water fluxes by extremes, different season lengths depending on ecoregion and latitude etc.
* * *

---

## Author Comment (AC1) · 31 Dec 2020

Referee #1 [General comment] Sun et al., performed a comprehensive evaluation of the nutrient-enabled version of the ORCHIDEE model (ORCHIDEE-CNP). The evaluations were made for biosphere carbon fluxes, N and P cycles, leaf and soil conditions and plant resource use efficiencies. Based on this, the authors were able to provide clear recommendations for future development. The extensive set of observational data that the authors use, together with the evaluation of different metrics is very powerful. The work provides a complete picture of the model performance. My main concern, however, is that due to the comprehensiveness of the work it is at the same time difficult to grasp the main messages when reading the results and discussion section. When I read the manuscript for the first time I was overwhelmed by all the

data and comparisons. I did not get a clear picture of the key messages in between the discussion of all the different metrics. The length of the manuscript contributes to this as well (27 pages of text, 18 figures and a table, plus several supplementary documents). Instead of showing comparisons of all available observational datasets, the authors could choose to only show those figures that help to illustrate their conclusion, and move other comparisons to the supplementary material and discuss them only shortly. For example, the authors discuss in sect 5.1.2 that it is difficult to falsify one model version over another based on the comparison of the de-trended anomalies in Fig 4, due to uncertainties in the observed GPP. As this figure did not contribute much to their final conclusion, the authors could choose to leave Fig. 4 out of the main manuscript. I would advise the authors to have a critical look at their figures, and the messages that they convey, to see which figures are really key to bring their message forward in order to condense (and thereby improve) the manuscript. Moreover, section 2 "Model description" gives an overview of the modifications that were made compared to ORCHIDEE-CNP v1.1, but it does not provide an overview of the nutrient flows of N and P. A brief description of steps in the N and P cycles would be helpful to understand the processes in this nutrient-enabled model version of ORCHIDEE. When such processes are introduced this would also help to understand the evaluation with observations later on. A discussion of Fig. 1 could serve this, as it is a nice illustration of such processes, but this figure is currently hardly discussed in the text. [Response] Thank you very much for your careful review and the positive comments with regard to our manuscript. We have revised the manuscript according to your suggestions and hope the readability of the manuscript has improved substantially due to focusing on key aspects. According to your suggestion on making the main text 'more clear and concise', we reconstructed the Result and Discussion sections with focusing on the evaluations for 4 key nutrient-related emerging properties of the model simulations which are linked to ecosystem gas exchanges and carbon storage: (1) vegetation resource use efficiencies, (2) the response of GPP to increasing $CO_2$, (3) ecosystem N and P turnover and openness, and (4) large-scale pattern of ecosystem stoichiometries. Point (1) and (2) control the response of vegetation carbon uptake which operates on timescales of years to decades, while point (3) and (4) control the response of the ecosystem carbon storage potential which operates on timescales of centuries and longer. This has been stated in the revised introduction section (Page 3; Lines 101-109). The choice was further based on the availability of observational data. To do so the following changes have been made which condensed the main text to 20 pages and 10 figures. First, information on processes which underlie the 4 key properties was moved to a large part to the SI. For example, we moved the detailed evaluation of single nutrient fluxes (e.g. BNF, P leaching etc.) and budgets into the SI (Sect. S2-S7 in supplement) which are now cited in the main text in the discussion of patterns in ecosystem nutrient turnover. Second, we condensed the information which was previously shown in three figures into one single figure by focusing on two statistical indexes (i.e. $R^2$ and relative mean square error) between model and reference datasets (Fig. 10). The original figures which contain additional information (spatial patterns, temporal evolution) are now shown in the SI. According to your suggestion on providing the overview of nutrient flows in the 'Model description', we added the brief description of steps in the N and P cycles for the important N and P fluxes: 'ORCHIDEE-CNP simulates the cycles of C, N and P which are described in detail elsewhere (Krinner et al., 2005; Zaehle and Friend 2010; Goll et al., 2014, 2017a, 2018). We here give a brief overview. P enters the ecosystem by release from minerals into the soil solution, whereas N is biologically fixed from an ample reservoir of dinitrogen. Dissolved nutrients are either taken up by vegetation, converted into soil organic matter or absorbed onto soil particles. Losses occur as leaching of dissolved nutrients, gaseous soil N emissions, or occlusion of P in secondary minerals. When nutrients are taken up by vegetation they are either stored internally or used to build new plant tissue driven by the availability of C, N and P in vegetation. The nutrient concentration of plant tissue varies within a prescribed range depending on the relative availability of C, N and P. Before plant tissue is shed, depending on the tissue a fixed fraction of the nutrients is recycled. The nutrients contained in dead plant tissue and organic matter are mineralized and released back into the soil solution.' (Pages 3-4; Lines 106-117). All of the specific comments and suggestions have been addressed and implemented in this revised manuscript. Responses to the specific comments can be found below. [Comment 1] P2, Line 49: "this direction of future carbon storage", what direction? [Response to #1] It indicates the overestimation in C storage. This sentence has been changed as: "Empirical stoichiometry observations were applied in the posteriori estimates of future carbon storage from land surface models (LSMs) lacking an explicit simulation of N and P biogeochemistry, which led consistently to an overestimation of future carbon storage in LSMs (Hungate et al., 2003; Wang and Houlton, 2009; Zaehle et al., 2015; Wieder et al., 2015)." (Page 2; Lines 46-50). [Comment 2] P2, line 70: "should look for" = needs. [Response to #2] We replaced "should look for" by "needs" according to your suggestion. [Comment 3] P2-3, line 75-81: Shorten or break up this sentence for readability. [Response to #3] According to your suggestion, we broke up this sentence as: "The evaluation for N and P together with carbon cycling in global LSMs remains very limited (Wang et al., 2010; Goll et al., 2012) but recent advances in ground-based measurements, ecological datasets and process understanding have made a better evaluation of C, N, P models feasible. The available nutrient datasets have allowed for meta-analyses of site-level nutrient fertilization experiments (e.g. Yuan and Chen, 2015; Wright, 2019), data-driven assimilation schemes to constrain nutrient budgets (Wang et al., 2018), new knowledge about the critical P-processes of sorption (Helfenstein et al., 2018; 2020) and phosphatase-mediated mineralization (Sun et al., 2020), global datasets of leaf nutrient content (Butler et al., 2017), and empirical constraints on the $CO_2$ fertilization effect on land carbon storage (Terrer et al., 2019; Liu et al., 2019)." (Pages 2-3; Lines 73-81). [Comment 4] P4, line 118: give full name of SOM. [Response to #4] The full name of SOM "soil organic matter" was given in this sentence. [Comment 5] P4, line 140-142: mention here the resolution of the ORCHIDEE run (0.5 degree) that differs from the 2.0 degree for ORCHIDEE-CNP. Otherwise this is only mentioned in the caption of Fig. 3, but it is relevant information. [Response to #5] According to your suggestion, we added one sentence to state the information of spatial resolution for ORCHIDEE as: "To disentangle the effect of introducing nutrient cycles into ORCHIDEE, we performed the same simulation with ORCHIDEE (revision 5375) which has no nutrient cycles and a comparable parameterization for other processes. ORCHIDEE was run at a higher spatial resolution (0.5ox0.5o) than ORCHIDEE-CNP. Prior to the analysis, the data from ORCHIDEE was remapped to the resolution of ORCHIDEE-CNP." (Page 5; Lines 155-159). [Comment 6] P8, line 277-279: check correctness of this sentence. [Response to #6] This sentence was corrected and removed into SI according to our re-constructions. The revised sentence is: "Thus, the annual soil P loss via surface runoff (kg P km-2 yr-1) from the ORCHIDEE-CNP output were extracted, and were compared with the GlobalNEWS2 load rates (ðİŠČload)." (Sect. S5 in the supplement). [Comment 7] P8, line 287: its uncertainties were calculated [Response to #7] We corrected it as "its uncertainties were calculated". [Comment 8] P8, line 290: based on [Response to #8] We corrected it as "based on". [Comment 9] P8, line 291: remove "speaking" [Response to #9] We removed the word "speaking" from this sentence. [Comment 10] P9, line 311: twice higher = twice as high [Response to #10] This sentence was deleted according to our re-constructions. [Comment 11] P9, line 316-318: The markers in Fig. 2b for temperate and western Europe don't seem to show this? [Response to #11] Thanks for pointing this out. We revised the text to be in consistent with the figure (Fig. S6 in the revised manuscript) as: "ORCHIDEE-CNP simulated comparable GPP values for most parts of the globe (Fig. S6a), and comparable NPP values for most of northern high-latitudes (Fig. S6b), which lie within the range given by the data-driven products." (Page 13; Lines 446-448). [Comment 12] P10, line345- 347: Over the whole range, Eco2 seems to be quite similar for ORCHIDEE-CNP and ORCHIDEE. Is the ORCHIDEE Eco2 higher only because it has more data points in the lower GPP range where Eco2 is clearly higher than those based on Campbell et al. (2017) and Ehlers et al. (2015)? What is the role of the different resolution of ORCHIDEE-CNP and ORCHIDEE here? Does the resolution explain that there are data points in the ORCHIDEE plot below âĹij400 GPP-296 and in between âĹij300-1000 GPP-396, that are not there for ORCHIDEE-CNP? [Response

to #12] Thanks for this question. We are lack of the simulation for ORCHIDEE with spatial resolution of 2o x 2o, which make it is hard to separate the role of different resolution. To keep the consistency of comparison, we resampled the ORCHIDEE outputs to 2o x 2o resolution before comparing the ECO2. This information is also provided in the main text: "All the gridded datasets with high spatial resolutions (Table 1) were resampled to the 2o x 2o resolution of the model output using area-weighted mean methods." (Page 7; Lines 261-262). Then, we found that ORCHIDEE still overestimated the Eco2 for low GPP region. Besides, the spatial pattern of ECO2 (Fig. S5) shows that ORCHIDEE simulate a much more higher value than ORCHIDEE-CNP in northern high latitudes. [Comment 13] P11, line 379: refer to Fig. 6a here already [Response to #13] This sentence has been deleted in the revised manuscript. [Comment 14] P11, line 381: "... but is close to JENA-inversion estimate during this period". Can you give the value for the CTracker atmospheric inversion? [Response to #14] This sentence has been deleted in the revised manuscript. To make the comparison for NBP clearer, we show the matrix of statistic indexes between ORCHIDEE-CNP and inversion data and mean value across Trendy ensemble (v6) (see the Response to the General comment) in the main text (Fig. 10). [Comment 15] P11, line 381: Global simulated NBP from ORCHIDEE-CNP. . .. [Response to #15] This sentence has been deleted in the revised manuscript (see Response to #14). The statement for the evaluation on NBP was revised as: "Net biome productivity (NBP) is defined as the net C exchange between the atmosphere and the terrestrial biosphere, that is the sum of net primary productivity, heterotrophic respiration and emissions due to disturbances; positive values denoting a land carbon sink. Compared to the three sets of atmospheric inversions (CAMS, JENA and CTracker), ORCHIDEE(-CNP) performs slightly worse than the mean of predictions from 16 land surface models from Trendy ensembles (v6) (Fig. 10c). ORCHIDEE-CNP shows a worse performance in inter-annual variability of NBP than ORCHIDEE when compared against inversion datasets at global scale and for the Northern Hemisphere. However, ORCHIDEE-CNP improved the performance of inter-annual variability of NBP against inversion datasets relative to ORCHIDEE for

tropical region (higher R2 and lower rMSE) with closer or even better fitness against inversion datasets than the mean value of Trendy ensemble models (Fig. 10c)." (Page 13; Lines 463-472). [Comment 16] P11, line 392: avoid the word "cause" here. Better say something like: "Therefore, the underestimation of the global C sink in ORCHIDEE-CNP during the last decades is primarily due to a lower C sink in the NH." [Response to #16] This sentence has been deleted in the revised manuscript (see Response to #14 and #15). [Comment 17] P12, line 410: abbreviation BNF was introduced already earlier. [Response to #17] We removed the full name of BNF here. [Comment 18] P13, line 457-468: this discussion of literature values is longer than needed, please shorten. [Response to #18] According to your suggestion, we removed this part of result for BNF to the SI (Sect. S4 in the supplement) and cited it in the section '5.3 Ecosystem N and P turnover openness'. [Comment 19] P15, line 537-538: it is not entirely clear to me how I can see the net N accumulation of 51.5 Tg N yr-1 from Fig 14a? [Response to #19] This part was removed to the SI and this sentence was deleted. [Comment 20] Discussion: as there were so many results presented, it is important to reference to a figure or paragraph from the results section that evidences your statements. E.g. in line 631-634; 635-638. [Response to #20] Thanks for this useful suggestion. We added the reference for the figures from the results rather than only citing the section number. [Comment 21] P18, line 643: "all models", you mean biosphere models? [Response to #21] We replaced the "models" by "LSMs". The revised sentence is: "The strength of the fertilization effect on GPP differs strongly between LSMs (Friedlingstein et al., 2014)." (Page 15; Lines 534-549). [Comment 22] P19, line 676-677: but aren't ORCHIDEE-CNP and ORCHIDEE forced with the same meteorology, and thus the same temperature, precipitation and radiation? [Response to #22] ORCHIDEE-CNP and ORCHIDEE used the same forcing data of meteorology from CRU-JRA-55. But BESS and MTE used climate datasets from CRU-NCEP. For analyzing the sensitivity of GPP anomaly to climates, we used CRU-JRA-55 for ORCHIDEE-CNP and ORCHIDEE and CRU-NCEP for BESS and MTE. This information has been added in the supplement S1I. [Comment 23] P20, line 691: you mean Fig. S7? [Response to #23]

It corresponds to Fig. S10 after we reshuffled the Supplementary. [Comment 24] P20, line 711: "...of the land C sink...", you mean "the size of the land C sink"? [Response to #24] This sentence has been revised as "Current LSM unanimously conclude that CO2 fertilization is the main driver of the land carbon sink and its trend (Friedlingstein et al., 2014), but it remains unclear to what extent other drivers (i.e. climate change, land management, nutrient deposition) contribute to the sink as well." (Page 18; Lines 659-661). [Comment 25] P21, line 768: is "capital" the right word here? Maybe use "pools"? [Response to #25] This sentence has been deleted in the revised manuscript. [Comment 26] P23, line 825: "due to that. . ." = "that is because . . .. " [Response to #26] This sentence was removed into SI. We revised it as "that is because". [Comment 27] P23, line 831: model = models [Response to #27] This sentence was removed into SI. We revised it as "LSMs". [Comment 28] P23, line 830-834: shorten this sentence for readability. [Response to #28] This sentence was removed into SI. According to your suggestion, we broke up this sentence as: "In the P-enabled LSMs, inorganic P processes operating on longer timescale (occlusion, strong sorption) are only simply represented (Wang et al., 2010; Yang et al., 2014; Goll et al., 2017b). This processes in LSMs is primarily based on calibration rather than data driven, which remain a large source of uncertainty regarding changes in P availability under elevated CO2 (Goll et al., 2012)." (Sect. S6 in the supplement). [Comment 29] Appendices and Supplementary Material: It is confusing that the manuscript contains Appendices ánd two supplementary documents. It would be good if all such supplementary material is combined into a single document. [Response to #29] According to your suggestion, we combined the Appendices into Supplementary Material. [Comment 30] Tables and Figures: Table 1: CMAS inversion = CAMS inversion? Fig 3, legend: Rose lines = Pink lines? [Response to #30] We are very sorry for the wrong spelling. The text in the legend and Table 1 has been revised to be "CAMS inversion". We corrected the legend of Fig. 5 (original Fig.3) as 'pink lines' instead of 'rose lines'. [Comment 31] Fig 7, complex, consider removing it to the supplementary material. [Response to #31] This figure is very important to illustrate the simulated global C, N and P fluxes and

storages by ORCHIDEE-CNP compared to GOLUM-CNP which is a data-driven modeling of steady-state of C, N and P dynamic. We simplified this figure (Fig. 2) and kept it in the main text. [Comment 32] Fig 10, Besides mentioning "model B" and "model C", also give the reference in the legend (like for model A, Peng et al., 2019 is mentioned). [Response to #32] According to your suggestion, we added the reference 'Cleveland et al., 1999' and 'Wang and Houlton, 2009' in the legend (Fig. S16). [Comment 33] Fig 12, 16, what is red and what is black? Fig 12, is n from the model the number of grid cells with that soil type? Fig 9, 12, 16, 18 add in the legend what the width of the bars indicates. Fig 16, what do a, b, c in the figures mean? [Response to #33] We revised the legends of those figures to make it clear to understand.

---

## Author Comment (AC2) · 31 Dec 2020

Referee 2 The work by Sun et al. is impressive in the sense that many analyses are taken to understand the Orchidee-CNP model. But on the other hand, the paper lacks a clear flow of arguments. [Comment 1] The reason why every time the models need to be more detailed is that we are not satisfied by the performance of the old models. If we only focus on carbon and water (WUE) then we clearly see problems in the dynamics, sinks and sources, which was the reason to include Nitrogen and now also the Phosphorus cycle. However, if one the main conclusions is that the current version of this model is unable to simulate carbon sinks, then the choices of expanding the model need to N and P should be much more discussed in detail. As long as we cannot model the carbon cycle, what kind of implications has this on the N and P cycle? If there is a

large problem in land carbon sink estimates, then I would like to see the consequences to all other coupled processes, including water. If this is large, then this should be solved first or we should simply conclude that there is too less understanding to couple the models as proposed in a global model. An evaluation as proposed by the authors of this version of the model doesn't help us in answering this problem. [Response to 1] Thanks for this comment. We agree that including more details (e.g. nutrients) into models are done with the aim to reach a better model performance (this was described on Page 2 Lines 69-71). One of the aims of LSM is the quantification of the response of the land C balance to man-made environmental changes in the past, present and future. Here we show that the inclusion of nutrient cycles in ORCHIDEE-CNP did deteriorate the simulated land C sink in the Northern Hemisphere (NH) for recent decades compared to ORCHIDEE. However, we do show that ORCHIDEE-CNP performs better in simulating the underlying mechanisms than ORCHIDEE: e.g. sensitivities of ecosystem productivity to increasing $CO_2$ and to variation in water availability. Moreover, ORCHIDEE-CNP tends to better reproduce observed vegetation resource use efficiencies (see further details below). This suggests that ORCHIDEE 'got the right result for the wrong reason' or, in other words, both models 'cannot model the C cycle'. However, ORCHIDEE and ORCHIDEE-CNP perform well compared to other LSM as indicated by the iLAMB benchmarking tool (Friedlingstein et al., 2019). Thus, they reflect our current capabilities in modelling the carbon cycle. Compared to ORCHIDEE, ORCHIDEE-CNP tends to improve the performance of resource use efficiencies, the sensitivity of plant productivity to increasing $CO_2$ ($CO_2$ fertilization effect), inter-annual variation of GPP in the northern hemisphere (NH). Besides, ORCHIDEE-CNP is able to better reproduce the variation of NBP in tropical regions. For NH, ORCHIDEE-CNP simulated a more realistic $CO_2$ fertilization effect, which can be explained by nutrient effects on plant carbon uptake in line with theory and observations (e.g. Jiang et al., 2020). The underestimated carbon sink in NH points toward a driver of the NH sink which is not included in ORCHIDEE and ORCHIDEE-CNP, e.g. forest regrowth (Pugh et al., 2019). We also showed that ORCHIDEE-CNP underestimates P availability in

the NH, thus another explanation is that the NH sink in this study is too low because of too strong P limitations in this region. The detailed explanation for the underestimated NH C sink can be found in Sect. 5.5.3. More developments are needed to improve ORCHIDEE-CNP in the NH. We emphasized here that our evaluation for N and P together with C cycling goes well beyond the evaluation of other global and site scale CNP models (Wang et al., 2010; Yang et al., 2014; Goll et al., 2012; Fleischer et al., 2019). We argue that our study provides insights on the strengths and weaknesses of ORCHIDEE-CNP and thus allows us (1) to define model applications for which realistic predictions could be expected (e.g. tropics) and (2) to interpret the model behavior. We realized that by our efforts in identifying the underlying reasons for the model biases, the focus of our original manuscript was shifted towards simulating the land C sink, away from the intended focus on the evaluation of key aspects governing the coupling of the C cycle to nutrient cycles. In the revised manuscript, we strengthen the focus on the evaluation for 4 key emerging model properties related to nutrients: (1) vegetation resource use efficiencies, (2) the response of GPP to increasing CO2, (3) ecosystem N and P turnover and openness, and (4) large-scale pattern of ecosystem stoichiometries (see details in Response to 2 of referee 2). The evaluation of land C sink serves as an example of the implication including nutrients in a major area of LSM application (i.e. dynamics of land C balance). In the revised paper, we reconstructed the result and discussion sections using this storyline to clarify the statements for pros and cons of ORCHIDEE-CNP as well as the ways to address the model biases. The main focus of this paper is the nutrients-related emerging model properties and an implication on C cycle. Considering the paper is very comprehensive, we choose to follow the storyline mentioned above and not include additional evaluation for water or energy fluxes. We will focus in more detail on the effects of nutrients on water and energy fluxes in our follow-up studies. [Comment 2] A second major concern is therefore that the evaluation is far too broad while missing the in-depth analyses. The number of figures are too many and jumping from one type of comparisons to another: a. On one hand, you are showing the dynamics, but then I would like to have much more

information on understanding the drivers. For instance, how much dependent is the dynamics of P and N on the P and N deposition? There are studies who have shown in other models that this N Deposition is one of the main drivers. b. Then you make some snap shots of global patterns, while later on you focus on different ecoregions and then different soil types and then on vegetation with different photosynthetic pathways. It would be very helpful to structure this far better and to integrate those aspects. [Response to 2] Thanks for this comment. An issue we face with the evaluation of global models is the limited availability of data for evaluation covering only a small subset of simulated variables. In addition, the temporal and spatial coverage of the scarce data varies among the datasets. To improve the structure of the presentation of our analysis, we focus in the revised manuscript on key aspects for the coupling of the cycles of C, N and P (see previous answers) and moved a large part of the analysis to the SI. To better 'integrate the various information', we focus now on 4 key aspects related to the effects of nutrients on the simulated response of ecosystem productivity and carbon storage to (changes in) climate and CO2: (1) vegetation resource use efficiencies, (2) the response of GPP to increasing CO2, (3) ecosystem N and P turnover and openness, and (4) large-scale pattern of ecosystem stoichiometries. Point (1) and (2) control the response of vegetation carbon uptake which operates on timescales of years to decades, while point (3) and (4) control the response of the ecosystem carbon storage potential which operates on timescales of centuries and longer. This has been stated in the revised introduction section (Page 3; Lines 101-109). The choice was further based on the availability of observational data. We agree that the depth of our analysis has been obscured in the original manuscript due to the various aspects we have analyzed and which have not been linked well enough in the discussion. We hope that by the narrower focus of the revised manuscript on key factors, we are now able to provide a 'depth analyses' of such effects. Our analysis does not aim at disentangling the drivers of the land carbon sink. The land sink part is merely an example for the implications of including nutrient cycling in ORCHIDEE (as stated now in the introduction Lines 108-109). We argue that the identification of underlying drivers (e.g.

CO2, nutrient deposition etc.) is out of the scope, in particular as there exists large uncertainty with respect to reconstructions of phosphorus deposition and the sensitivity of ecosystem carbon storage to nutrient deposition (Wang et al., 2017) which requires a study on its own. [Comment 3] A third major concern is the for me random way of comparing the results. I found the comparison with only ORHIDEE-C not very convincing. Why not comparing to the average performance of the land models as done in TRENDY? There are other global model results as well on C and N. For instance, LPJ guess. [Response to 3] Thanks for pointing this out. We want to stress that the results of ORCHIDEE-CNP are not only compared to ORHIDEE-C, but also compared to data-driven products and observations. The comparison to ORCHIDEE-C serves the purpose to separate the effect of including nutrient cycles on the simulated C cycle, as ORCHIDEE-CNP differs primarily from ORCHIDEE-C with respect of having cycles of N and P. This is not possible using results from other models as there are various additional differences making it nearly impossible to explain differences among models with certainty (e.g. Rogers et al., 2017). Instead of comparing our simulated NP flows to results from other land surface models as suggested by the referee, we compared them to results from the model-data-synthesis tool GOLUM-CNP which provides more robust estimates of CNP flows than a LSM. The structure of GOLUM-CNP allows the assimilation of observational data from various sources, which is not possible in LSM due to their complexity. Nonetheless, we use results from the Trendy model ensemble (i.e. the ensemble average which has been demonstrated to perform substantially better than any single model for various aspects) for the land C balance (as for example done in Global Carbon Project; Friedlingstein et al., 2019). [Comment 4] L402: why do you have a smaller natural land cover? Is it a problem from ORCHIDEE or from GOLUM-CNP. Is it then still useful to compare? [Response to 4] Thanks for this question. ORCHIDEE-CNP uses a different land cover than the one used to upscale results from the biome-scale model GOLUM-CNP. The main difference in land cover originates from the omission of managed land area in GOLUM-CNP. As the biome-scale model GOLUM-CNP does not resolve spatial variation within a biome, we compared the results on biome scale for nutrients use efficiencies and ecosystem N and P residence time. When comparing the global spatial pattern of ecosystem nutrients openness, we masked the areas with managed lands (agricultural and pasture lands) >50For the comparison of C, N and P flows and storages with GOLUM-CNP, we only compared the values per area (unit: g C/N/P m-2) and removed the panel with global values (unit: Pg C/N/P) in the revised manuscript (Sect. 4.1). We admit that the comparisons on a per area basis for some highly sensitive variables to agricultural activities (e.g. P leaching) are not real valid. Thus, we stated this issue (Page 5; Lines 328-335) and did additional comparisons for those variables using more valid datasets that considering agricultural activities (Sect. S5 in the supplement). [Comment 5] Comparing global news N-leaching: other forcings → this doesn't help us in understanding the role of the different mechanisms → can you also compare it with similar forcings? If not, is it still valid to include this comparison? [Response to 5] Thanks for this question. This study is the first among all current P-enabled LSMs studies to evaluate the N and P loads from land to rivers on both basin and global scale. Despite the different forcings used, ORCHIDEE-CNP simulated N and P loads from land to rivers in the same order of magnitude as the GlobalNEWS2 model. After we reconstructed the storyline in the revised paper (see Response to 1 and 2), all of the information for N and P leaching went into the SI (Sect. S5 in the supplement) and serves as the explanation for biased nutrients turnover and openness (Sect. 4.4 and 5.3). Although we agree that the investigations for N and P leaching drivers are valuable, we argue that this is out of the scope of this study. The main purpose of our evaluation is to provide a whole picture of the current states of C, N and P dynamics by ORCHIDEE-CNP rather than to understand the drivers of changes the different mechanisms for specific process (also see Response to 2). [Comment 6] How did you downscale from HYDE3.2 to 1x1 km? Did you use the same allocation rules as done by Klein Goldewijk for the 30 minute resolution? [Response to 6] We did not downscale HYDE3.2 to finer resolution. In contrast, we aggregated it to coarser resolution of 2o × 2o to constrain our land-cover maps. We corrected the description of the historical land-cover maps for ORCHIDEE-CNP as

follows (also in the Sect. 3.1.3 of the revised manuscript; Page 5; Lines 169-183). "The historic land-cover change maps were based on the European Space Agency Climate Change Initiative (ESA-CCI) land-cover data (Bontemps et al., 2013). To be used by global vegetation models ORCHIDEE-CNP, ESA-CCI land-cover data were aggregated to 2o × 2o, and grouped into PFTs using the reclassification method from Poulter et al. (2011, 2015). The fraction of cropland and pasture in the PFT map was further constrained by the cropland area and the sum of pasture and rangeland area of the year 2010 in the History Database of the Global Environment land use data set (HYDE 3.2; Klein Goldewijk et al., 2017a, b) respectively, which were also aggregated to 2o × 2o. The above processes produced a reference ESA-CCI-based PFT map for the year 2010. The land-use changes derived from and Land-Use Harmonization (LUH) v2 (http://luh.umd.edu/data.shtml; an update release of Hurtt et al., 2011) were aggregated to 2o × 2o and then were applied to this reference PFT map to constrain the land-cover changes of forest, grassland, pasture and rangeland, and cropland during the period 1700-2017 using the backward natural land cover reconstruction method of Peng et al. (2017). As a result, a set of historic PFT maps suitable for global vegetation models were established distinguishing global land-cover changes for the period of 1700-2017 at 2o × 2o resolution." [Comment 7] L890: N and P leaching: the current problem of understanding N and P leaching is large: there are all kind of confounding factors that determine these leaching rates which are in the end extremely important to understand water quality and functioning of the system. In the current paper I cannot find this sensitivity back. Leaching is highly dynamic due to fire, soil water fluxes by extremes, different season lengths depending on ecoregion and latitude etc. [Response to 7] As we argued before, understanding the role of the underlying mechanisms for N and P leaching is out of the scope of this study (see Response to 2 and 5). Besides, we cannot explore the sensitivity of N and P leaching to fire, extreme events or erosion as they are currently not resolved in ORCHIDEE-CNP. References Friedlingstein, P., Jones, M., O'Sullivan, M., Andrew, R., Hauck, J., Peters, G., Peters, W., Pongratz, J., Sitch, S., and Le Quéré, C.: Global carbon budget 2019, Earth Syst.

Sci. Data, 11, 1783-1838, http://doi.org/10.5194/essd-11-1783-2019, 2019. Goll, D. S., Brovkin, V., Parida, B., Reick, C. H., Kattge, J., Reich, P. B., Van Bodegom, P., and Niinemets, Ü.: Nutrient limitation reduces land carbon uptake in simulations with a model of combined carbon, nitrogen and phosphorus cycling, Biogeosciences, 9, 3547-3569, http://doi.org/10.5194/bg-9-3547-2012, 2012. Jiang, M., Medlyn, B. E., Drake, J. E., Duursma, R. A., Anderson, I. C., Barton, C. V., Boer, M. M., Carrillo, Y., Castañeda-Gómez, L., Collins, L., Crous, K. Y., De Kauwe, M. G., dos Santos, B. M., Emmerson, K. M., Facey, S. L., Gherlenda, A. N., Gimeno, T. E., Hasegawa, S., Johnson, S. N., Kännaste, A., Macdonald, C. A., Mahmud, K., Moore, B. D., Nazaries, L., Neilson, E. H. J., Nielsen, U. N., Niinemets, Ü., Noh, N. J., Ochoa-Hueso, R., Pathare, V. S., Pendall, E., Pihlblad, J., Piñeiro, J., Powell, J. R., Power, S. A., Reich, P. B., Renchon, A. A., Riegler, M., Rinnan, R., Rymer, P. D., Salomón, R. L., Singh, B. K., Smith, B., Tjoelker, M. G., Walker, J. K. M., Wujeska-Klause, A., Yang, J., Zaehle, S. and Ellsworth, D. S.: The fate of carbon in a mature forest under carbon dioxide enrichment, Nature, 580(7802), 227–231, http://doi.org/10.1038/s41586-020-2128-9, 2020. Pugh, T. A., Lindeskog, M., Smith, B., Poulter, B., Arneth, A., Haverd, V., and Calle, L.: Role of forest regrowth in global carbon sink dynamics, Proc. Natl. Acad. Sci. U. S. A., 116, 4382-4387, http://doi.org/10.1073/pnas.1810512116, 2019. Rogers, A., Medlyn, B. E., Dukes, J. S., Bonan, G., von Caemmerer, S., Dietze, M. C., Kattge, J., Leakey, A. D. B., Mercado, L. M., Niinemets, Ü., Prentice, I. C., Serbin, S. P., Sitch, S., Way, D. A., and Zaehle, S.: A roadmap for improving the representation of photosynthesis in earth system models, New Phytol., 213, 22– 42, 2017. Wang, R., Goll, D., Balkanski, Y., Hauglustaine, D., Boucher, O., Ciais, P., Janssens, I., Penuelas, J., Guenet, B., and Sardans, J.: Global forest carbon uptake due to nitrogen and phosphorus deposition from 1850 to 2100, Glob. Change Biol., 23, 4854-4872, http://doi.org/10.1111/gcb.13766, 2017. Wang, Y., Law, R., and Pak, B.: A global model of carbon, nitrogen and phosphorus cycles for the 1610 terrestrial biosphere, Biogeosciences, 7, http://doi.org/10.5194/bg-7-2261-2010, 2010. Yang, X., Thornton, P., Ricciuto, D., and Post, W.: The role of phosphorus dynamics in tropical forests- -a modeling study using CLM-CNP, Biogeosciences, 11,

http://doi.org/10.5194/bg-11-1667-2014, 2014.

---

## Author Comment (AC3) · 31 Dec 2020

Please see the attached pdf.

Please also note the supplement to this comment:
https://gmd.copernicus.org/preprints/gmd-2020-93/gmd-2020-93-AC3-supplement.pdf
* * *

---

## Author Response (AR2)

**[Comment from the Referee]** *The authors made a great effort to make a selection of their main results and combined figures (e.g. Fig 10) to make a more condensed manuscript. Still, I find the discussion section rather lengthy, and it is hard to find the main messages of the work without reading the text in detail. One suggestion for readability is to change the section titles to more descriptive titles that give a hint to the key message of each section.*

**[Author response]** Thank you very much for your review and the helpful suggestion on improving the readability of the manuscript. According to your suggestion, we used the more descriptive titles for the Discussion section, which are listed below:

'5. Discussion

   5.1 Inclusion of nutrient cycling improves use efficiencies of other plant resources

   5.2 Inclusion of nutrient cycling improves $CO_2$ fertilization effect

   5.3 Ecosystem nutrient turnover and openness indicates model biases in boreal phosphorus availability

   5.4 Model biases in stoichiometry indicate need for refinement of process representation

   5.5. Nutrient effects on carbon cycling

      5.5.1 Inclusion of nutrient cycling improves the inter-annual variability of GPP

      5.5.2 Inclusion of nutrient cycling deteriorates phenology and seasonality of GPP

      5.5.3 Inclusion of nutrient cycling leads to an underestimation of the land carbon sink
'

Besides, the titles for Sect. 2-4 were revised to be more clear and concise as:

'2. Modelling

   2.1 Model description

   2.2 Simulation setup

      2.2.1 Meteorological data

      2.2.2 Land cover

      2.2.3 Soil and lithology datasets

      2.2.4 Atmospheric nitrogen and phosphorus deposition

      2.2.5 Nutrient management

3. Evaluation

   3.1 Ecosystem productivity

   3.2 Resource use efficiencies

4. Results

   4.1 Carbon, nitrogen and phosphorus flows and storages

,